# EXPONENTIAL LOW-RANK ADAPTERS

## ABSTRACT

Low rank adaptation (LoRA) is a standard parameter efficient fine tuning method, but its updates are rank limited and act as local additive perturbations of the weights. We introduce exponential low rank adapters (ELRA), which replace LoRA's additive update $\Delta W = AB$ with a multiplicative transformation $W_{\text{new}} = \exp(\eta AB) W_0$, where $A \in \mathbb{R}^{d \times r}$ and $B \in \mathbb{R}^{r \times d}$ define a low rank generator and $W_0$ is the frozen pretrained weight. The matrix exponential lifts the generator to a full rank, invertible map that acts coherently on $W_0$. Geometrically, ELRA traces curves on $GL(d)$; when the generator is normal, the path $t \mapsto \exp(tG)W_0$ is a constant speed, locally energy minimizing geodesic under the left invariant trace metric. This motivates ELRA-PSD, which constrains $G$ to be symmetric positive semidefinite, stabilizing training by enforcing geodesic flows with an energy control on path length. To couple stability with expressivity, we propose ELRA-Hyb, which interpolates between PSD and general generators, starting stable and progressively unlocking capacity. Experiments on diverse language and vision benchmarks show that ELRA-Hyb consistently outperforms state of the art LoRA variants under matched parameter budgets. Code is available at https://anonymous.4open.science/r/ELRA-7629/README.md.

## 1 INTRODUCTION

Low-rank adaptation (LoRA) (Hu et al., 2022) has become a popular parameter-efficient fine-tuning (PEFT) technique for large language models. LoRA freezes the original model weights $W_0 \in \mathbb{R}^{d \times k}$ and injects trainable low-rank matrices $A \in \mathbb{R}^{d \times r}$ and $B \in \mathbb{R}^{r \times k}$ into each layer's weight update (so the weight change $\Delta W = AB$). This drastically reduces the number of trainable parameters while maintaining good performance on many tasks.

Despite its practicality, LoRA's expressiveness is bounded by the rank $r$. When the target update is intrinsically high-rank, a fixed low-rank factorization may underfit relative to full fine-tuning (Huang et al., 2025). Increasing $r$ often helps but trades away parameter savings, which undermines the very purpose of PEFT. This motivates a central question: how can we retain parameter efficiency while overcoming the rank bottleneck?

Several recent methods aim to boost expressiveness primarily by enlarging the attainable rank of the update. RandLoRA (Albert et al.) composes learned coefficients with a bank of random bases to emulate full-rank changes with few learned parameters, but the fixed, non-learnable bases can limit task specificity. HiRA (Huang et al., 2025) leverages elementwise interactions between pretrained weights and low-rank factors to lift the effective rank, yet its dependence on the structure of $W_0$ constrains flexibility. These limits point to a complementary strategy: keep the parameter budget fixed and improve expressiveness by reparameterizing the update itself rather than increasing rank.

Following the above-mentioned reparameterization strategy, we introduce **e**xponential **l**ow **r**ank **a**dapters (ELRA), which replace the additive update by a multiplicative transformation,

$$W_{\text{new}} = \exp(\eta AB) W_0,$$

where $A \in \mathbb{R}^{d \times r}$ and $B \in \mathbb{R}^{r \times d}$ parameterize a low-rank generator $G = \eta AB$, and $\exp(\cdot)$ denotes the ***matrix*** exponential which is fundamentally different from an elementwise exponential that acts independently on each entry. The matrix exponential lifts this low-rank generator to a full-rank, invertible map that acts coherently across all dimensions of $W_0$.

From a geometric viewpoint, ELRA traces curves on the orbit $GL(d) \cdot W_0 = \{EW_0 : E \in GL(d)\}$. When the generator is *normal*, the path $t \mapsto \exp(tG)W_0$ has constant speed and is a locally

energy-minimizing geodesic under the left-invariant trace metric. This motivates *ELRA-PSD*, which constrains $G$ to the symmetric positive semidefinite form $G = \eta AA^\top$. The resulting geodesic updates improve stability, while an energy penalty controls path length.

To couple stability with expressivity, we introduce *ELRA-Hyb*, which interpolates between PSD and general generators: training begins in the PSD regime for stability, then progressively shifts toward the full generator to increase capacity.

Lastly, it is important to contrast our work with LieRA (Si et al., 2025), which applies an ***elementwise*** exponential to a low-rank generator and modulates $W_0$ through the Hadamard product $W_{\text{LieRA}} = W_0 \circ \exp(\Delta)$. This operation provides fine-grained, per-parameter scaling of $W_0$ but does not introduce new linear combinations across feature channels, since each weight is modified independently. In other words, the adapter preserves the connectivity pattern of $W_0$ and does not perform global feature mixing or basis transformations. In contrast, ELRA applies a left action in $\mathrm{GL}(d)$ through $W_{\text{ELRA}} = \exp(\eta AB), W_0$, which enables dense, invertible transformations (e.g., rotations, shears, anisotropic scalings) of the entire feature space.

In summary, the contributions of the paper are as follows:

- We propose ELRA, a geometry aware PEFT method that applies the *matrix* exponential of a low rank generator to the frozen weights, producing a full rank update while keeping a LoRA level parameter budget. We also derive a Woodbury style (Woodbury, 1950) reduction that evaluates $\exp(\eta AB)W_0$ via small $r \times r$ matrices, keeping the computational cost close to LoRA (Proposition 1).

- We analyze ELRA and prove key properties that explain its performance (Propositions 2 to 5).

- Leveraging ELRA's geodesic structure, we introduce two variants: ELRA-PSD and ELRA-Hyb.

- We conduct extensive experiments and analyses across language and vision benchmarks, demonstrating consistent gains over strong LoRA variants under matched parameter budgets.

**Notation**  Scalars use lowercase $a$, vectors bold lowercase $\boldsymbol{a}$, and matrices uppercase $A$. We write $\|\cdot\|_2$ for the spectral norm, $\|\cdot\|_{\text{F}}$ for the Frobenius norm, $\text{tr}(\cdot)$ for trace, $\text{rank}(\cdot)$ for rank, and $\exp(\cdot)$ for the matrix exponential. For a square $M$, $\lambda_{\min}(M)$ and $\lambda_{\max}(M)$ are its extremal eigenvalues; for any $M$, $\sigma_{\min}(M)$ and $\sigma_{\max}(M)$ are its extremal singular values and $\kappa_2(M) = \sigma_{\max}(M)/\sigma_{\min}(M)$. Finally, $GL(d)$ is the group of invertible $d \times d$ matrices and $\mathfrak{gl}(d)$ its Lie algebra.

## 2 RELATED WORK

**Low rank adaptation and its extensions.**  LoRA (Hu et al., 2022) is the standard PEFT baseline. Many works aim to raise expressiveness without abandoning parameter efficiency. LoRA-XS (Bałazy et al., 2024) restricts updates to a truncated SVD subspace to reduce parameters, though the fixed basis can limit adaptability. DoRA (Liu et al., 2024) decomposes weights into magnitude and direction and applies low rank updates to the direction, improving optimization but not the attainable rank. ABBA (Zhang et al., 2024) composes two low rank adapters via elementwise multiplication to gain capacity, at the cost of doubled adapter complexity. LoRAN (Li et al., 2024) introduces nonlinearity inside the low rank update, which can improve approximation but complicates tuning and deployment. MoRA (Xu et al., 2024) replaces the two factor LoRA decomposition with a square matrix sandwiched by fixed projections, enabling higher effective rank but introducing design choices for the projections. Yen et al. (2025) introduce adaptive matrix preconditioning method to achieve transformation invariance.

**Initialization and merging strategies.**  PiSSA (Meng et al., 2024) initializes adapters from dominant singular directions of pretrained weights, MiLoRA (Wang et al., 2025a) uses the least dominant directions, and LoRA-GA (Wang et al., 2024) uses singular features of gradients. Jiang et al. (2025) reduces redundancies before merging adapters back into the base model, and He et al. (2025) selects sparse submatrices to narrow the gap between PEFT and full fine tuning.

**Closest to our approach.**  HiRA (Huang et al., 2025) increases expressiveness through Hadamard interactions between $W_0$ and low-rank factors, and LieRA (Si et al., 2025) applies elementwise exponentials of a low-rank matrix to rescale individual entries. Ji et al. (Ji et al., 2025) propose

sine-activated low-rank matrices, where a sinusoidal nonlinearity is applied elementwise to a low-rank product (and, in the PEFT setting, to LoRA/DoRA adapters) to increase the effective rank without increasing the number of trainable parameters. All of these approaches act coordinatewise on the entries of $W_0$ or on the adapter, preserving the connectivity pattern and zero structure and not inducing coupled row–column transformations such as rotations, shears, or global basis changes in feature space. In contrast, ELRA acts on the full general linear group $GL(d)$ via a matrix exponential, producing dense, invertible transformations that couple all coordinates (a direct experimental comparison with HiRA and LieRA appears in Appendix G).

## 3 METHODOLOGY

### 3.1 MOTIVATION

LoRA restricts updates to additive perturbations of rank $r$, leaving many directions unreachable when $r$ is small. We seek a mechanism that keeps LoRA's small parameter footprint, yields full rank transformations, and couples all coordinates of $W_0$. A natural route is to work in the general linear group $\mathrm{GL}(d)$ and use a *matrix* exponential to map a low-rank generator to a full-rank transformation.

### 3.2 EXPONENTIAL LOW-RANK ADAPTERS (ELRA)

Given a frozen weight matrix $W_0 \in \mathbb{R}^{d \times k}$, we introduce two trainable low-rank matrices

$$A \in \mathbb{R}^{d \times r}, \quad B \in \mathbb{R}^{r \times d}, \tag{1}$$

along with a *trainable* positive scaling factor $\eta > 0$. We define the low-rank generator:

$$G = \eta\, AB \in \mathbb{R}^{d \times d}. \tag{2}$$

The updated weight is then defined via a matrix exponential acting on the left[1]:

$$W_{\text{new}} = \exp(G)\, W_0. \tag{3}$$

Note that unlike conventional LoRA where $B \in \mathbb{R}^{r \times k}$, here we have $B \in \mathbb{R}^{r \times d}$. This is because the matrix exponential $\exp(G)$ is only defined for square matrices.

**Remark 1.** Compared to standard LoRA, which uses $A \in \mathbb{R}^{d \times r}$, $B \in \mathbb{R}^{r \times k}$ for a total of $r(d + k)$ parameters, our square-generator formulation uses $B \in \mathbb{R}^{r \times d}$, resulting in $2rd$ trainable parameters. The difference per layer is $r(k - d)$, meaning: ($i$) for $k > d$, our method uses fewer parameters; ($ii$) for $k < d$, it uses more. In transformers, FFN expand and compress layers appear in pairs, so these differences cancel out, leaving the total trainable count essentially ***unchanged***[2].

During the training phase, gradients are backpropagated through the matrix exponential using automatic differentiation. Only the adapter parameters $A$, $B$, and $\eta$ receive updates.

Next, to ensure that the computational cost of ELRA remains comparable to conventional LoRA, we introduce a proposition based on a Woodbury-style identity (Woodbury, 1950). This result provides an efficient way to evaluate the exponential update without ever forming the full matrix $e^G$.

**Proposition 1.** Define $\phi_1(Z) \triangleq \sum_{n \geq 1} \frac{Z^{n-1}}{n!}$. Then

$$e^{\eta AB} = I + \eta\, A\, \phi_1(\eta BA)\, B, \tag{4}$$

and for any $W \in \mathbb{R}^{d \times k}$, we have $e^{\eta AB} W = W + \eta\, A\, \phi_1(\eta BA)\, (BW)$.

Proofs for all statements in this section appear in Appendix C. To apply $(e^{\eta AB} W_0)$ to input $\boldsymbol{x}$:

$$\boldsymbol{u} = W_0 \boldsymbol{x}, \qquad \boldsymbol{v} = B\boldsymbol{u}, \qquad \boldsymbol{t} = \phi_1(\eta BA)\, \boldsymbol{v}, \qquad \boldsymbol{y} = \boldsymbol{u} + \eta\, A\boldsymbol{t}. \tag{5}$$

As such, only the small $r \times r$ matrix $BA$ enters the exponential evaluation.

*Cost.* Forming $BA$ costs $\mathcal{O}(dr^2)$ once per layer, evaluating $\phi_1(\eta BA)$ costs $\mathcal{O}(r^3)$, and applying the update to a matrix $W \in \mathbb{R}^{d \times k}$ costs $\mathcal{O}(drk)$. For $r \ll d$, the dominant term is $\mathcal{O}(drk)$, which is of the same order as LoRA and much lower than the $\mathcal{O}(d^2k)$ cost of full-rank updates.

---

[1]A more general double-sided variant, $W_{\text{new}} = \exp(L)W_0 \exp(R)$, is analyzed in Appendix B. Empirically, the left-exponential update achieves superior performance.

[2]This holds for most transformer LLMs, for example GPT, LlaMA, Mistral or Mixtral, Gemma, and Phi.

### 3.3 BASIC PROPERTIES OF ELRA

In this subsection, we study the properties of ELRA. First, Proposition 2 states that ELRA recovers LoRA update in the first-order approximation, and strictly extends it through higher order terms.

**Proposition 2.** Let $W_{\mathrm{new}} = \exp(\eta AB)W_0$. Then for sufficiently small $\eta$, we have

$$W_{\mathrm{new}} = W_0 + \eta A(BW_0) + O(\eta^2). \tag{6}$$

Next, Proposition 3 states that unlike the conventional LoRA which can possibly reduce the rank by up to its adapter rank (see Appendix D), ELRA preserves the rank of $W_0$.

**Proposition 3.** $\exp(G)$ is invertible with $(\exp(G))^{-1} = \exp(-G)$ and $\det(\exp(G)) = \exp(\mathrm{tr}\, G) \neq 0$. Consequently,

$$\mathrm{rank}(W_{\mathrm{new}}) = \mathrm{rank}(W_0), \qquad \mathcal{N}(W_{\mathrm{new}}) = \mathcal{N}(W_0), \qquad \mathrm{Col}(W_{\mathrm{new}}) = \exp(G)\, \mathrm{Col}(W_0). \tag{7}$$

**Corollary 1.** Since pretrained weights are typically full rank in practice, we may assume that $W_0$ is full rank. In particular, if $W_0 \in \mathbb{R}^{d \times k}$ has $\mathrm{rank}(W_0) = \min(d, k)$, then $W_{\mathrm{new}}$ is also full rank for any generator $G = \eta AB$.

We further analyze the ELRA's singular values in the sequel.

**Proposition 4.** Let $H = \frac{1}{2}(G + G^\top)$ and $W_{\mathrm{new}} = \exp(G)W_0$. Then

$$\sigma_{\min}(W_{\mathrm{new}}) \geq \sigma_{\min}(\exp(G))\, \sigma_{\min}(W_0) \geq \exp\big(\lambda_{\min}(H)\big)\, \sigma_{\min}(W_0). \tag{8}$$

In particular, ELRA's update cannot drive a nonzero singular value of $W_0$ to zero.

The next result controls the overall conditioning of the layer after an ELRA step.

**Proposition 5.** Let $G \in \mathbb{R}^{d \times d}$, $H = \frac{1}{2}(G + G^\top)$, and $W_{\mathrm{new}} = \exp(G)W_0$. Then

$$\kappa_2(W_{\mathrm{new}}) \leq \exp\big(\lambda_{\max}(H) - \lambda_{\min}(H)\big)\, \kappa_2(W_0). \tag{9}$$

As per Proposition 5, the exponential left action cannot arbitrarily worsen conditioning, the condition number of $W_{\mathrm{new}}$ is at most $\exp(\lambda_{\max}(H) - \lambda_{\min}(H))$ times that of $W_0$, so choosing generators with small spectral spread (for example via modest step size or regularization) keeps conditioning stable.

## 4 GEODESIC STRUCTURE OF ELRA

ELRA admits a natural interpretation via matrix group geometry. In Section 4.1, we examine the properties of the exponential map that form the foundation of our formulation. Building on these insights, Section 4.2 introduces two practical variants, ELRA-PSD and ELRA-Hyb.

### 4.1 WHY ELRA UPDATE IS SPECIAL

On the general linear group $GL(d)$ equipped with the left-invariant trace metric

$$g_E(\xi, \zeta) = \mathrm{tr}\big[(E^{-1}\xi)^\top (E^{-1}\zeta)\big], \tag{10}$$

the *body velocity* of a smooth curve $E(t) \in GL(d)$ is

$$\Omega(t) = E(t)^{-1}\dot{E}(t) \in \mathfrak{gl}(d), \qquad \text{where} \quad \dot{E}(t) \triangleq \frac{d}{dt}E(t). \tag{11}$$

For this metric, the Euler-Arnold equation reads (Arnol'd, 2013)

$$\dot{\Omega} = -\mathrm{ad}_\Omega^\dagger \Omega, \qquad \text{with} \quad \mathrm{ad}_X^\dagger Y = X^\top Y - YX^\top. \tag{12}$$

Equivalently,

$$\dot{\Omega} = -[\Omega^\top, \Omega], \qquad \text{where} \quad [X^\top, X] \triangleq X^\top X - XX^\top. \tag{13}$$

A sufficient condition that makes the exponential path a geodesic is that the body velocity be constant and *normal*. Specifically, if $\Omega(t) \equiv X$ and $[X^\top, X] = 0$ (that is, $X^\top X = XX^\top$), then the right-hand side of Equation (13) vanishes, and since $\Omega(t)$ is constant we have $\dot{\Omega} = 0$, so Equation (12)

is satisfied. Because $\Omega = E^{-1}\dot{E} \equiv X$, we obtain the linear matrix ordinary differential equation $\dot{E} = EX$ with solution

$$E(t) \; = \; E(0)\, e^{tX} \; = \; E_0 \exp(tX). \tag{14}$$

Hence $E(t)$ is a geodesic in $GL(d)$ under the left-invariant trace metric, and it has constant body speed $\|\Omega\| = \|X\|$ with respect to the Frobenius norm induced by the metric, where $\|X\|^2 = \mathrm{tr}(X^\top X)$.

One-parameter subgroups have the general form $E(t) = E_0 \exp(tX)$, but they are not geodesics for this metric in general. The normality condition above is a sufficient criterion that ensures geodesicity and constant body speed.

On a Lie group, any $C^1$ one-parameter subgroup has the form $E(t) = E_0 \exp(tX)$ for some $X \in \mathfrak{gl}(d)$; along this curve the body velocity is constant, $\Omega(t) \equiv X$. Hence the exponential gives the unique path through $E_0$ with fixed body generator $X$. Under the left-invariant trace metric on $GL(d)$, if $X$ is normal, this exponential path satisfies the Euler-Arnold equation and is therefore a geodesic; moreover, geodesics are locally length (energy) minimizing for sufficiently short times.

With this setup, we now state the geodesic and energy properties of the ELRA path.

**Proposition 6.** Let $W_0 \in \mathbb{R}^{d \times k}$ have full row rank and let

$$G = \eta AB, \qquad A \in \mathbb{R}^{d \times r},\ B \in \mathbb{R}^{r \times d},\ \eta \in \mathbb{R}.$$

Assume $G$ is *normal*, i.e., $[G^\top, G] = 0$. Define $F_{\mathrm{ELRA}}(t) = \exp(tG)\, W_0$, $t \in [0, 1]$. Then:

($i$) The curve $t \mapsto \exp(tG)$ is a geodesic in $GL(d)$ endowed with the left-invariant trace metric. It has constant body velocity, and

$$\|\dot{F}_{\mathrm{ELRA}}(t)\|^2_{g^{\mathrm{orb}}} = \mathrm{tr}(G^\top G) \quad \text{for all } t. \tag{15}$$

($ii$) Since $W_0$ has full row rank, the stabilizer $\{E \in GL(d) : EW_0 = W_0\}$ is trivial, hence $\Phi : GL(d) \to GL(d) \cdot W_0$, $\Phi(E) = EW_0$ is a diffeomorphism. Define the orbit metric by

$$g^{\mathrm{orb}}_{EW_0}(UW_0, VW_0) \triangleq g_E(U, V). \tag{16}$$

With this choice $\Phi$ is an isometry, so $F_{\mathrm{ELRA}}$ is a geodesic in $GL(d) \cdot W_0$. Moreover, it *locally* minimizes the Riemannian energy

$$\mathcal{E}[F] = \tfrac{1}{2} \int_0^1 g^{\mathrm{orb}}_{F(t)}\big(\dot{F}(t), \dot{F}(t)\big)\, dt \tag{17}$$

among $C^1$ curves in $GL(d) \cdot W_0$ with the same endpoints.

### 4.2 From geodesics to practical variants: ELRA-PSD and ELRA-Hyb

The geodesic result implies that when the generator $G$ is normal, the exponential action $F(t) = \exp(tG)W_0$ traces a constant-speed, locally energy-minimizing path on the orbit $GL(d) \cdot W_0$. This perspective highlights not only stability but also exploration: unlike the additive update $W_0 + AB$, which can follow irregular trajectories, the exponential path evolves with *uniform* speed, ensuring smoother and more balanced traversal of the weight space.

In addition, constant speed avoids sudden jumps that destabilize training, while energy minimization ensures that additional epochs simply extend the same smooth trajectory. As a result, convergence is typically reached in fewer epochs compared to other methods. Building on these insights, we introduce two practical variants of ELRA, namely ELRA-PSD and ELRA-Hyb.

#### 4.2.1 ELRA-PSD

**Geometric motivation.** For a PSD generator $G = \eta AA^\top$, the exponential map $\exp(tG)$ traces a geodesic in $GL(d)$ under the left-invariant trace metric. Along this path, the speed and energy are constant and equal to $\frac{1}{2}\mathrm{tr}(G^\top G)$, which gives us direct control of the update magnitude through the Frobenius norm of $AA^\top$. The PSD structure also guarantees that $\exp(G)$ preserves or increases all singular values of $W_0$ (see Proposition 7), preventing harmful spectral collapse.

**Training objective.** In ELRA-PSD, we restrict the generator to

$$G = \eta\, AA^\top, \qquad A \in \mathbb{R}^{d \times r},$$

and update the layer by

$$W_{\text{new}} = \exp(G)\, W_0.$$

The training loss becomes

$$\mathcal{L}_{\text{psd}} = \mathcal{L}_{\text{task}} + \lambda\, \text{tr}(G^\top G),$$

where the regularizer corresponds exactly to the squared geodesic energy of the path $t \mapsto \exp(tG)$ and enforces controlled, stable updates.

**Conditioning and stability.** From Proposition 5, a PSD generator satisfies

$$\kappa_2(W_{\text{new}}) \;\leq\; \exp\big(\eta\, \lambda_{\max}(AA^\top)\big)\, \kappa_2(W_0),$$

meaning the condition number can grow only by a bounded, well-controlled factor. Since $\exp(G)$ cannot decrease singular values and cannot collapse small directions, ELRA-PSD preserves the spectral structure of $W_0$ and maintains stable conditioning during early training.

**Proposition 7.** If $G = \eta\, AA^\top$ with $\eta > 0$, then

$$\sigma_{\min}\big(\exp(\eta AA^\top)W_0\big) \;\geq\; \exp\big(\eta\lambda_{\min}(AA^\top)\big)\, \sigma_{\min}(W_0) \;\geq\; \sigma_{\min}(W_0). \tag{18}$$

Thus ELRA-PSD never decreases the smallest singular value, even when $AA^\top$ is rank deficient.

**Remark 2.** Although ELRA-PSD paths are geodesics that locally minimize Riemannian energy given fixed endpoints, in practice the endpoints are not fixed during training. The optimizer can increase the generator norm $\|G\|$ arbitrarily, which yields valid geodesics but with excessively long trajectories and unstable updates. The energy penalty $\lambda\, \text{tr}(G^\top G)$ does not alter the geodesic property, but instead biases the solution toward shorter geodesics, thereby stabilizing training, analogous to choosing a ten meter straight line over a ten thousand kilometer one.

### 4.2.2 ELRA-Hyb

ELRA-PSD stabilizes training by constraining the generator to be PSD, but this symmetry reduces expressivity. A general generator $G = \eta AB$ is more expressive yet less stable early in training. To balance these, *ELRA-Hyb* interpolates between PSD and general forms, starting with stable geodesic flows and gradually allowing richer dynamics. Mathematically, let $\alpha \in [0, 1]$ be a smooth, monotonically decreasing schedule. We define the following generator

$$G \triangleq \eta\Big(\alpha\, AA^\top \;+\; (1 - \alpha)\, AB\Big). \tag{19}$$

Hence, in ELRA-Hyb, we optimize a single loss with the energy of the current generator:

$$\mathcal{L}_{\text{hyb}} \;=\; \mathcal{L}_{\text{task}} \;+\; \lambda\, \|G\|_F^2 \;=\; \mathcal{L}_{\text{task}} \;+\; \lambda\, \text{tr}(G^\top G). \tag{20}$$

In practice, we choose exponential decay for $\alpha$ starting from 1 to 0. Note that when $\alpha = 1$, $\mathcal{L}_{\text{hyb}}$ reduces to $\mathcal{L}_{\text{psd}}$, and when $\alpha = 0$, we get the original ELRA introduced in Section 3. To compute $\text{tr}(G^\top G) = \|G\|_F^2$ efficiently, we use the cyclic property of the trace: by setting $S = A^\top A \in \mathbb{R}^{r \times r}$ and $C = BB^\top \in \mathbb{R}^{r \times r}$,

$$\|G\|_F^2 = \eta^2\Big(\alpha^2\, \|AA^\top\|_F^2 + (1 - \alpha)^2\, \|AB\|_F^2 + 2\,\alpha(1 - \alpha)\, \langle AA^\top,\, AB \rangle_F\Big), \tag{21a}$$

$$\|AA^\top\|_F^2 \;=\; \text{tr}\big((A^\top A)^2\big) \;=\; \text{tr}(S^2), \tag{21b}$$

$$\|AB\|_F^2 \;=\; \text{tr}\big((A^\top A)(BB^\top)\big) \;=\; \text{tr}(SC), \tag{21c}$$

$$\langle AA^\top,\, AB \rangle_F = \text{tr}\big((AA^\top)^\top(AB)\big) = \text{tr}\big(AA^\top AB\big) = \text{tr}\big((A^\top A)(BA)\big) = \text{tr}\big(S\,(BA)\big). \tag{21d}$$

All terms involve only the $r \times r$ Gram/cross-Gram matrices $S$, $C$, and the product $BA$, so no $d \times d$ matrices are formed; the cost is $\mathcal{O}(dr^2)$ in the low-rank regime. The overall procedure for ELRA-Hyb is summarized in Algorithm 1.

---

**Algorithm 1** Training Procedure for ELRA-Hyb.

---

1: **Input:** Frozen weight $W_0 \in \mathbb{R}^{d \times k}$, rank $r$, steps $T$, schedule $\alpha \in [0, 1]$, regularizer $\lambda$.
2: **Init:** $A \sim \mathcal{N}(0, \sigma^2) \in \mathbb{R}^{d \times r}$, $B \sim \mathcal{N}(0, \sigma^2) \in \mathbb{R}^{r \times d}$, $\eta > 0$.
3: **for** $t = 1, \dots, T$ **do**
4:     $C \leftarrow \eta(\alpha\, A^\top + (1-\alpha)\, B) \in \mathbb{R}^{r \times d}$       $\triangleright\ G = A\,C$ with $G = \eta(\alpha A A^\top + (1-\alpha) A B)$
5:     $Z \leftarrow CA \in \mathbb{R}^{r \times r}$,    $M \leftarrow \phi_1(Z)$       $\triangleright$ Woodbury-style identity in Proposition 1
6:     **Forward:** for each minibatch input $\boldsymbol{x}$:
      $\boldsymbol{u} \leftarrow W_0 \boldsymbol{x} \in \mathbb{R}^d$,    $\boldsymbol{v} \leftarrow C\boldsymbol{u} \in \mathbb{R}^r$,    $\boldsymbol{t} \leftarrow M v \in \mathbb{R}^r$
      $\boldsymbol{y} \leftarrow \boldsymbol{u} + A\boldsymbol{t} \in \mathbb{R}^d$       $\triangleright\ \boldsymbol{y} = e^G(W_0 \boldsymbol{x})$
7:     Compute task loss $\mathcal{L}_{\text{task}}$ from $\boldsymbol{y}$.
8:     Compute $\|G\|_F^2$ from Equation (21).
9:     $\mathcal{L} \leftarrow \mathcal{L}_{\text{task}} + \lambda \|G\|_F^2$.
10:    Update $(A, B, \eta)$ by backprop.
11: **end for**
12: **Output:** Trained $A, B, \eta$.

---

## 5 EXPERIMENTS

We conduct a comprehensive evaluation of ELRA across multiple domains including natural language generation (Section 5.1) and understanding (Section 5.2), image classification (Section 5.3), and reasoning and question-answering benchmarks (Appendix F). For the baselines, we consider LoRA (Hu et al., 2022), RSLoRA (Kalajdzievski, 2023), DoRA (Liu et al., 2024), LoRA+ (Hayou et al., 2024), OLoRA (Büyükakyüz, 2024), PiSSA (Meng et al., 2024), LoRA-GA (Wang et al., 2024), AdaLoRA (Zhang et al., 2023), HiRA (Huang et al., 2025), LoRA-Pro (Wang et al., 2025b).

**Training setup.** To enable a fair comparison with prior work, we adopt an experimental protocol closely following that of LoRA-GA (Wang et al., 2024). Model fine-tuning is performed with the AdamW optimizer (Kingma, 2014), using $\beta_1 = 0.9$, $\beta_2 = 0.999$, and zero weight decay. A cosine learning-rate schedule with a 3% warm-up phase is applied. LoRA adapters are inserted into all linear layers except embeddings, normalization layers, and the final classification head.

For natural-language understanding benchmarks, we fine-tune a T5-base model (Raffel et al., 2020) with a learning rate of $1 \times 10^{-4}$, sequence length of 128, and batch size of 32. Dialogue generation, mathematical reasoning, and code generation experiments are run on LlaMA-2-7B (Touvron et al., 2023) with a learning rate of $2 \times 10^{-5}$, maximum context length of 1024 tokens, and an effective macro-batch size of 32. Image classification is performed using CLIP-ViT-B/16 (Radford et al., 2021), fine-tuned with a learning rate of $1 \times 10^{-4}$ and batch size of 64.

Unless otherwise noted, for all methods, the rank is set to $r = 8$. However, for ELRA-PSD the rank is $r = 16$ so that the number of trainable parameters are the same for all methods. In all experiments, we report the accuracy of **ELRA**, **ELRA-PSD**, and **ELRA-Hyb**, as well as ELRA-Hyb combined with LoRA-GA (Wang et al., 2024), denoted by **ELRA-Hyb$_{\text{GA}}$** which leverages singular features of gradients to initialize low-rank weights. Additional experimental details are provided in Appendix H.

In all tables, best and second best results are **bold** and underlined, respectively. Scores are averaged over *three* random seeds, and the corresponding standard deviations are reported in Appendix J.

### 5.1 RESULTS ON NATURAL LANGUAGE GENERATION TASKS

• **Backbones.** We conduct experiments on natural language generation tasks using two backbone models: Llama-3.1-8B-Base (Dubey et al., 2024) and Llama-2-7B-Base (Touvron et al., 2023), fine-tuned on dialogue, mathematical reasoning, and code generation datasets.

• **Evaluation Metric.** For dialogue generation, we use MT-Bench (Zheng et al., 2023) and report dialogue-level scores (0-10) as judged by strong LLM evaluators, including GPT-4o (Achiam et al., 2023), Gemini-1.5-Pro (Team et al., 2024), and Llama-3.1-70B-Instruct (Dubey et al., 2024), following the official MT-Bench prompt setup. For mathematical reasoning, we evaluate on GSM8K (Cobbe

Table 1: Performance of fine-tuning methods on Llama-3.1-8B-Base and Llama-2-7B-Base across MTBench, GSM8K, and HumanEval.

| Method | Llama-3.1-8B-Base | | | Llama-2-7B-Base | | |
|---|---|---|---|---|---|---|
| | MTBench | GSM8K | HumanEval | MTBench | GSM8K | HumanEval |
| Full-FT | 5.88 | 73.69 | 51.63 | 5.30 | 59.36 | 35.31 |
| LoRA ICLR 2022 | 6.15 | 67.78 | 43.09 | 5.61 | 42.08 | 14.76 |
| RSLoRA arXiv 2023 | 6.18 | 68.36 | 45.78 | 5.25 | 45.62 | 16.01 |
| DoRA ICML 2024 | 6.24 | 69.17 | 43.70 | 5.97 | 53.07 | 19.75 |
| LoRA+ ICML 2024 | 6.35 | 71.29 | 44.51 | 5.71 | 52.11 | 18.17 |
| OLoRA arXiv 2024 | 6.13 | 68.54 | 43.29 | 5.30 | 43.29 | 17.22 |
| PiSSA NeurIPS 2024 | 6.08 | 68.56 | 44.10 | 5.30 | 44.54 | 16.02 |
| LoRA-GA NeurIPS 2024 | 5.99 | 71.39 | 43.29 | 5.95 | 53.60 | 19.81 |
| AdaLoRA ICLR 2023 | 6.19 | 70.63 | 41.46 | 5.57 | 50.72 | 17.80 |
| HiRA ICLR 2025 | 6.28 | 73.29 | 45.52 | 5.71 | 54.51 | 19.24 |
| LoRA-Pro ICLR 2025b | 6.30 | **75.49** | 47.25 | 5.72 | 57.57 | 22.97 |
| **ELRA** | 6.21 | 68.21 | 44.21 | 5.71 | 44.21 | 16.73 |
| **ELRA-PSD** | 6.26 | 71.47 | 45.67 | 5.90 | 52.47 | 20.44 |
| **ELRA-Hyb** | **6.42** | 75.25 | 48.31 | **6.01** | 57.90 | 24.18 |
| **ELRA-Hyb$_{GA}$** | 6.40 | 75.42 | **48.77** | 5.95 | **57.93** | **24.53** |

et al., 2021) and report accuracy based on exact-match regular expressions. For code generation, we use HumanEval (Chen et al., 2021) and measure functional correctness with the PASS@1 metric.

• **Results.** Across Llama-3.1-8B and Llama-2-7B (Table 1), ELRA variants are better than strong adapter baselines in almost all settings. ELRA-Hyb attains the best MTBench scores (6.42; 6.01), and ELRA-Hyb$_{GA}$ achieves the strongest HumanEval (48.77; 24.53), further narrowing the gap to full fine-tuning. On GSM8K, ELRA-Hyb$_{GA}$ is competitive on Llama-3.1-8B (75.42; LoRA-Pro 75.49) and sets the adapter SOTA on Llama-2-7B (57.93). The original ELRA and ELRA-PSD also consistently beat vanilla LoRA. For training time and parameter budget analysis, refer to Appendix I.

## 5.2 RESULTS ON NATURAL LANGUAGE UNDERSTANDING TASKS

• **Backbones.** We fine-tune the T5-Base model (Raffel et al., 2020) on five GLUE benchmarks (Wang et al.), including MNLI, SST-2, CoLA, QNLI, and MRPC.

• **Results.** As shown in Table 2, ELRA-Hyb and ELRA-Hyb$_{GA}$ achieve the highest overall average scores of 88.20 and 88.45, respectively, surpassing full-FT (87.91) and all other baselines.

Table 2: Performance of fine-tuning T5-Base on 5 sub-tasks of the GLUE benchmark.

| Method | MNLI | SST-2 | CoLA | QNLI | MRPC | Avg |
|---|---|---|---|---|---|---|
| Full-FT | 86.33 | 94.75 | 80.70 | 93.19 | 84.56 | 87.91 |
| LoRA | 85.30 | 94.04 | 69.35 | 92.96 | 68.38 | 82.08 |
| RSLoRA | 85.73 | 94.19 | 72.32 | 93.12 | 52.86 | 79.64 |
| DoRA | 85.67 | 94.04 | 72.04 | 93.04 | 68.08 | 82.57 |
| LoRA+ | 85.81 | 93.85 | 77.53 | 93.14 | 74.43 | 84.95 |
| PiSSA | 85.75 | 94.07 | 74.27 | 93.15 | 76.31 | 84.71 |
| LoRA-GA | 85.70 | 94.11 | 80.57 | 93.18 | 85.29 | 87.77 |
| AdaLoRA | 85.45 | 93.69 | 69.16 | 91.66 | 68.14 | 81.62 |
| **ELRA** | 85.72 | 94.07 | 73.21 | 93.09 | 74.86 | 84.19 |
| **ELRA-PSD** | 85.80 | 94.22 | 76.61 | 93.34 | 84.13 | 86.82 |
| **ELRA-Hyb** | 85.90 | 94.25 | 80.77 | **93.43** | 86.65 | 88.20 |
| **ELRA-Hyb$_{GA}$** | **85.97** | **94.52** | **81.53** | 93.42 | **86.83** | **88.45** |

## 5.3 RESULTS ON IMAGE CLASSIFICATION TASKS

• **Backbones.** We fine-tune CLIP-ViT-B/16 across seven benchmarks: Stanford Cars (Krause et al., 2013), DTD (Cimpoi et al., 2014), EuroSAT (Helber et al., 2019), GTSRB (Houben et al., 2013), RESISC45 (Cheng et al., 2017), SUN397 (Xiao et al., 2010), and SVHN (Netzer et al., 2011).

• **Results.** The results are reported in Table 3. ELRA-Hyb and ELRA-Hyb$_{GA}$ achieve the highest accuracy across all seven datasets. Specifically, the average accuracy is even higher than Full-FT.

## 5.4 GEOMETRIC STABILITY DIAGNOSTICS: PER-STEP SPEED AND SPIKES

Here, we assess update smoothness on DTD dataset via per-step speed and we count "spike events" in speed, and we also examine the training loss and gradient norm.

Table 3: Performance of fine-tuning CLIP-ViT-B/16 on 7 image classification tasks.

| Method | Cars | DTD | EuroSAT | GTSRB | RESISC45 | SUN397 | SVHN | *Avg* |
|---|---|---|---|---|---|---|---|---|
| Zero-shot | 63.75 | 44.39 | 42.22 | 35.22 | 56.46 | 62.56 | 15.53 | 45.73 |
| Full-FT | 84.23 | 77.44 | 98.09 | 94.31 | 93.95 | 75.35 | 93.04 | 88.06 |
| LoRA | 72.81 | 73.92 | 96.93 | 92.40 | 90.03 | 70.12 | 88.02 | 83.46 |
| DoRA | 73.72 | 73.72 | 96.95 | 92.38 | 90.03 | 70.20 | 88.23 | 83.48 |
| LoRA+ | 72.87 | 73.96 | 97.01 | 92.42 | 89.96 | 70.17 | 88.08 | 83.51 |
| RSLoRA | 82.38 | 78.03 | 98.06 | 95.04 | 93.96 | 75.38 | 92.74 | 87.94 |
| RandLoRA | 83.23 | 77.31 | 98.05 | 94.24 | 94.37 | 74.19 | 92.97 | 87.77 |
| LoRA-Pro | 85.87 | 78.64 | 98.46 | 95.66 | 94.75 | 76.42 | 94.63 | 89.20 |
| LoRA-GA | 85.18 | 77.50 | 98.05 | 95.28 | 94.43 | 75.44 | 93.68 | 88.51 |
| **ELRA** | 73.15 | 74.17 | 97.20 | 93.85 | 91.37 | 71.63 | 90.05 | 84.49 |
| **ELRA-PSD** | 78.33 | 75.15 | 97.50 | 94.19 | 93.06 | 74.17 | 93.40 | 86.54 |
| **ELRA-Hyb** | 85.92 | 78.43 | **98.62** | **96.51** | **95.01** | 76.83 | **95.32** | 89.52 |
| **ELRA-Hyb$_{GA}$** | **86.07** | **78.87** | 98.53 | 96.37 | 94.81 | **76.87** | 95.22 | **89.53** |

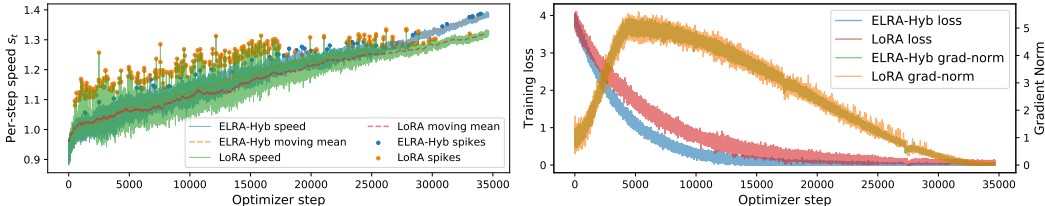

Figure 1: *Left*: per-step speed $s_t$ with moving mean and $3\sigma$ spike markers; *Right*: training loss and adapter gradient norm over steps, with smoother, more stable trajectories for ELRA-Hyb.

**per-step speed.** For each adapter $\ell = 1, \ldots, L$ let $G_t^{(\ell)}$ be the generator at optimizer step $t$ (for LoRA, $G_t^{(\ell)} = \eta_t A_t^{(\ell)} B_t^{(\ell)}$; for ELRA-Hyb, its current hybrid generator). Define $\Delta G_t^{(\ell)} \triangleq G_t^{(\ell)} - G_{t-1}^{(\ell)}$, and $s_t \triangleq \left( \sum_{\ell=1}^{L} \|\Delta G_t^{(\ell)}\|_F^2 \right)^{1/2}$. We plot $s_t$ with a causal moving mean (window $w = 200$). A spike is declared when $s_t > \mu_t + 3\sigma_t$, where $\mu_t, \sigma_t$ are the trailing window mean and standard deviation. ELRA-Hyb shows far fewer spikes

Table 4: Ablation on $\lambda$ for ELRA-Hyb on CARS, DTD, and EUROSAT.

| ELRA-Hyb | Cars | DTD | EuroSAT |
|---|---|---|---|
| $\lambda=0$ | 85.02 | 77.35 | 98.21 |
| $\lambda=1\times10^{-6}$ | 85.41 | 78.09 | 98.45 |
| $\lambda=5\times10^{-6}$ | 85.77 | 78.24 | 98.58 |
| $\lambda=1\times10^{-5}$ | 85.92 | 78.43 | 98.62 |
| $\lambda=3\times10^{-5}$ | 85.62 | 78.41 | 98.57 |
| $\lambda=1\times10^{-4}$ | 84.82 | 77.84 | 98.35 |

than LoRA in Figure 1 (left). In addition, we measure the coefficient of variation, calculated as $\text{CV}(s) = \frac{\text{std}(s_t)}{\max\{\text{mean}(s_t),\epsilon\}}$, with $\epsilon = 10^{-12}$, which is lower for ELRA-Hyb (0.14) than LoRA (0.29).

**Loss and gradient norm.** We record the training loss $\ell_t$ and the global adapter gradient norm $g_t = \|\nabla_{\theta_{\text{adapter}}} \ell_t\|_2$. Curves in Figure 1 (right) complement the above stability diagnostics.

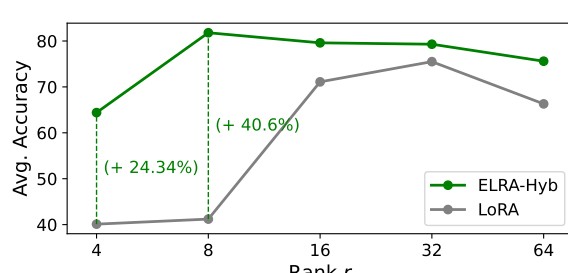

## 6 ABLATION STUDY

### 6.1 ABLATION ON $\lambda$

We ablate the coefficient $\lambda$ in ELRA-Hyb

Figure 2: Performance of ELRA-Hyb vs. LoRA at different ranks $r$. ELRA-Hyb shows clear gains, especially for small $r$.

(Table 4). Performance is flat for $\lambda \in [5\times10^{-6}, 3\times10^{-5}]$ with a peak near $1\times10^{-5}$; too little or too much regularization hurts. Strong scores at $\lambda=0$ are plausible since the PSD→general schedule $\alpha_t$ already stabilizes training; the energy term mainly shortens the path and damps excursions. Additional decay-parameter ablations are in Appendix K.

## 6.2 ABLATION ON RANK $r$

This section analyzes how varying the rank $r \in \{4, 8, 16, 32, 64\}$ affects the performance of ELRA-Hyb and LoRA when fine-tuning LLaMA-7B on commonsense reasoning benchmarks. Mean accuracies for each rank are presented in Figure 2. ELRA-Hyb consistently outperforms LoRA, with the largest differences at lower ranks. At $r = 4$, ELRA-Hyb achieves $64.44\%$, compared to LoRA's $40.10\%$. At $r = 8$, ELRA-Hyb reaches $81.83\%$, while LoRA remains at $41.23\%$. Although the gap narrows as rank increases, ELRA-Hyb maintains a consistent advantage, demonstrating greater robustness under aggressive rank reduction.

## 7 CONCLUSION

We presented ELRA, a multiplicative PEFT that applies the matrix exponential of a low rank generator to frozen weights, yielding full rank, invertible updates with a LoRA level parameter budget. The method admits a geometric view on $GL(d)$, and with normal generators the induced paths are constant speed geodesics. We introduced two practical variants: ELRA-PSD for stable symmetric generators with energy control, and ELRA-Hyb that begins in the PSD regime and gradually unlocks asymmetric capacity. Across language and vision benchmarks, ELRA-Hyb delivers consistent gains over strong LoRA baselines under matched parameter budgets.

## REPRODUCIBILITY STATEMENT

We have taken steps to ensure our results are reproducible. All model and algorithmic details, training procedures, hyperparameters, evaluation protocols, and metrics are specified. An anonymized GitHub repository contains the source code and configuration files, and pre-trained checkpoints. All datasets used in our experiments are publicly available.

## LLM USAGE STATEMENT

LLM used only for grammar and wording edits; no generation of ideas, methods, analyses, results, or citations. Authors reviewed all edits and accept full responsibility.

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

# A    RELATED WORK

## OTHER PEFT METHODS.

Other PEFT families modify either the input or the backbone. Prompt-based methods add trainable virtual tokens and optimize only those tokens. For example, Lester et al. (2021) propose Prompt Tuning, which learns task-specific tokens at the input layer, and Liu et al. (2022) extend this idea to multiple layers with P-Tuning. These methods add very few parameters but can be sensitive to initialization, and the inference cost grows with prompt length because transformer attention has quadratic complexity (Vaswani et al., 2017). Adapter-based methods instead insert small trainable modules into largely frozen networks. Houlsby et al. (2019) first introduced Adapters, and Mahabadi et al. (2021) proposed Compacter, which improves efficiency through hypercomplex low-rank layers. Parallel adapter variants (He et al., 2021) place these modules alongside the main path. While these methods are effective, they alter the forward graph during training and inference, which can introduce extra overhead compared with approaches that leave the backbone unchanged.

## LORA-XS (BAŁAZY ET AL., 2024): EXTREMELY SMALL ADAPTATION (SVD-BASED)

LoRA-XS aggressively reduces trainable parameters by applying truncated SVD to the weight matrix, freezing the resulting low-rank bases, and learning only a small $r \times r$ matrix within that subspace. This yields over 100× parameter reduction compared to LoRA while remaining competitive, making it ideal when extreme parameter efficiency is the priority.

**Why our method is better:** Unlike LoRA-XS, which fixes its update subspace and thus inherits LoRA's expressivity limits, our approach learns adaptation directions end-to-end. This allows us to represent a wider range of transformations and achieve higher fine-tuning quality on complex LLM tasks, trading a modest increase in parameters for significantly better performance.

## DORA (LIU ET AL., 2024): WEIGHT-DECOMPOSED LOW-RANK ADAPTATION

DoRA, developed by NVIDIA, decomposes each weight matrix into magnitude and direction, applying LoRA-style low-rank updates **only** to the direction. This design better matches full fine-tuning, which adjusts both components freely, whereas LoRA mixes them. By fine-tuning only directions and preserving magnitudes, DoRA achieves more stable and effective learning and consistently outperforms LoRA across LLMs, vision-language models, and diffusion models, without adding extra parameters.

**Why our method is better:** DoRA primarily improves *training dynamics* but still inherits LoRA's expressivity limit because updates remain low-rank. Our method removes this rank constraint, allowing full-rank updates and richer transformations. It is also simpler, requiring no weight decomposition or special update rules, making it an easy drop-in replacement that achieves higher performance when greater expressivity is needed.

## HIRA (HUANG ET AL., 2025): HADAMARD HIGH-RANK ADAPTATION

HiRA breaks LoRA's rank barrier by applying element-wise products with the base weight matrix. It keeps the usual low-rank $A, B$ matrices but computes the update as $\Delta W = (W \circ U)V$, where $\circ$ is the Hadamard product. This can produce an update of rank up to $r_1 \times r_2$, allowing HiRA to reach much higher effective rank than LoRA with the same adapter size. In practice, HiRA improves adaptability and outperforms LoRA across multiple tasks, confirming the benefit of higher-rank updates.

**Why our method is better:** HiRA's updates are tied to the original weight values, meaning it can only rescale existing weights rather than create entirely new patterns. If $W$ contains zeros or lacks structure needed for the target task, HiRA cannot introduce new nonzero entries or large rotations in weight space. Our method achieves full-rank updates without depending on $W$, allowing it to generate arbitrary weight deltas and capture transformations beyond simple rescaling. This broader update space leads to more robust adaptation and consistently better results, especially under domain shift or tasks requiring large structural changes.

### RANDLORA (ALBERT ET AL.): FULL-RANK ADAPTATION VIA RANDOM BASES

RandLoRA aims to overcome LoRA's rank limitation by enabling full-rank updates without training a full weight matrix. It constructs a set of fixed random matrices as bases and learns small diagonal scaling matrices to combine them. The resulting update is a linear combination of these random components, which can approximate any full-rank update while keeping the number of trainable parameters low. Experiments show that RandLoRA significantly reduces the gap between LoRA and full fine-tuning, and in some vision-language tasks even matches full fine-tuning performance.

**Why our method is better:** RandLoRA's random bases are not task-specific and may require many components to span a useful subspace. Its diagonal scaling further limits how flexibly bases can be combined, as each one is scaled uniformly. Our approach learns the basis directions directly, ensuring that all trainable parameters contribute to solving the task. This yields more targeted updates, potentially faster convergence, and higher accuracy, especially on tasks where random directions are insufficient.

### SINE-ACTIVATED LOW-RANK MATRICES (JI ET AL., 2025): RANK EXPANSION VIA ELEMENTWISE NONLINEARITIES

Ji et al. propose to enhance the expressivity of low-rank matrices by applying a sinusoidal nonlinearity elementwise to a low-rank product. Concretely, given a decomposition $UV^\top$, they form

$$\tilde{W} = \gamma \, \sin(\omega \, UV^\top),$$

where the sine activation is applied entrywise. The key insight is that this elementwise nonlinearity breaks the low-rank structure and can increase the effective rank of the adapter, enriching its singular spectrum without introducing additional trainable parameters. In the PEFT setting, they apply the same idea to LoRA and DoRA adapters, showing that sine-modulated low-rank adapters can achieve stronger performance than their linear counterparts. Their theoretical analysis focuses on rank expansion and spectral diversification of the transformed matrix, and the method is evaluated primarily against standard LoRA and DoRA on several vision and language tasks.

**Why our method is better:** Sine-activated low-rank adapters improve expressivity through an elementwise nonlinearity, but the resulting transformation remains coordinatewise. Each entry of $W_0$ (or of the adapter) is modulated independently, preserving the original connectivity pattern of the pretrained layer and preventing global mixing between feature channels. In contrast, ELRA applies a *matrix* exponential, generating a full-rank, invertible linear transformation in $GL(d)$ that couples all rows and columns of $W_0$. This enables rich global interactions (such as rotations, shears, and anisotropic scalings) that are not achievable through elementwise nonlinearities. ELRA also provides explicit geometric and spectral guarantees—including rank preservation, conditioning bounds, and a geodesic interpretation—that are absent in sine-activation methods. Finally, while Ji et al. compare mainly with LoRA and DoRA, our experiments include many stronger state-of-the-art baselines, offering a more rigorous demonstration of performance gains in competitive PEFT settings.

### LIERA (SI ET AL., 2025): ELEMENTWISE EXPONENTIAL ADAPTATION VIA ABELIAN LIE GROUPS

LieRA replaces additive LoRA updates with multiplicative ones grounded in Lie group theory. It treats weights as elements of an Abelian group under the Hadamard product and applies the exponential map from a low-rank generator: $W_{\text{new}} = W_0 \circ \exp(\Delta)$, where $\Delta$ is low-rank and $\circ$ is elementwise multiplication. This yields smooth, per-weight scaling with low parameter count and can reach full-rank effects when $W_0$ is full-rank. Empirically, LieRA improves over LoRA on vision and language benchmarks, particularly when fine-grained modulation helps.

**Why our method is better:** LieRA scales weights entrywise and cannot model global row or column interactions. ELRA applies a true matrix exponential to a low-rank generator, $W = \exp(\eta AB) W_0$, producing a full-rank, invertible linear transform over the entire weight space. This enables structured global updates such as rotations, basis changes, and feature mixing, including activation of previously zeroed features. ELRA retains smoothness and invertibility but operates in $GL(n)$, providing a richer class of transformations under similar parameter budgets.

ABBA (ZHANG ET AL., 2024): HADAMARD PRODUCT OF LOW-RANK MATRICES FOR HIGH EXPRESSIVITY

ABBA maximizes expressivity under a strict parameter budget by combining two independent low-rank updates via a Hadamard product: $\Delta W = (B_1 A_1) \circ (B_2 A_2)$. While each $B_i A_i$ is low-rank, their elementwise product can reach much higher effective rank, allowing ABBA to represent a richer set of transformations than a single LoRA module. Both updates are fully learnable and decoupled from $W$, enabling ABBA to capture entirely new patterns. The authors show theoretically that ABBA strictly contains LoRA's solution space for the same parameter count and report state-of-the-art results on reasoning and QA tasks.

**Why our method is better:** Our method achieves comparable expressivity with a simpler architecture that requires only one low-rank update, avoiding the complexity of managing two adapters and their interactions. ABBA doubles the number of learned matrices and needs specialized Khatri-Rao factorization for efficiency, whereas our approach works within standard LoRA implementations with minimal overhead. This simplicity makes it easier to tune and potentially more robust, while still matching or exceeding ABBA's performance.

LORAN (LI ET AL., 2024): LOW-RANK ADAPTATION WITH NON-LINEAR TRANSFORMATION

LoRAN increases expressivity by inserting a non-linear transformation into the LoRA update. Instead of a purely linear $\Delta W = BA$, it applies a function $f$ to obtain $\Delta W = f(BA)$, where $f$ is designed to warp the low-rank update into a better shape. This can approximate updates that would require higher rank under a linear model. Reported results on tasks such as SAMSum and 20 Newsgroups show consistent gains over LoRA, with notable improvements at very low ranks.

**Why our method is better:** LoRAN enhances expressivity through nonlinearity but adds complexity for optimization and deployment, since $f(BA)$ may not merge cleanly into base weights and introduces extra hyperparameters. Our method keeps a linear formulation with effectively unbounded rank, which preserves simple merging and inference while prov

MORA (XU ET AL., 2024): HIGH-RANK UPDATING WITH A SQUARE MATRIX

MoRA targets high-rank updates with limited parameters by replacing LoRA's two skinny factors with a trainable square matrix of size $r \times r$ and fixed in/out projections. Concretely, it learns $M \in \mathbb{R}^{r \times r}$ and uses fixed operators $P_{\text{in}} \in \mathbb{R}^{r \times d}$ and $P_{\text{out}} \in \mathbb{R}^{d \times r}$ to form $\Delta W = P_{\text{out}} M P_{\text{in}}$. Only $M$ is trained, so the parameter count is $r^2$, yet the update can reach rank $r$. After training, the effect merges into the base weights through the projections. Reported results show improvements over LoRA on memory-intensive settings and comparable performance elsewhere.

**Why our method is better:** Fixed projections can discard task-relevant information and require careful design, and selecting $r$ balances expressivity against parameter growth. Our method avoids these knobs by operating directly in the full space or a learned subspace without imposed bottlenecks. It keeps a standard adapter form, is easy to implement on top of LoRA frameworks, and provides higher expressivity across tasks, not only in memory-intensive scenarios.

## B  DOUBLE-SIDED EXTENSION AND ITS LIMITATIONS

A natural variant is to update on both sides of the weight,

$$W_{\text{new}} = \exp(L) W_0 \exp(R), \tag{22}$$

with $L \in \mathbb{R}^{d \times d}$ and $R \in \mathbb{R}^{k \times k}$. This acts on the double orbit $GL(d) \cdot W_0 \cdot GL(k)$. While appealing for its symmetry, it consistently underperforms the single sided form. The reason is over parameterization and gauge freedom: many pairs $(L, R)$ yield nearly the same $W_{\text{new}}$ because small moves along the left and right stabilizers of $W_0$ cancel, creating flat directions and poor conditioning. Optimizers then spend effort along redundant paths instead of producing effective changes.

In contrast, single sided ELRA moves on the left orbit $\{\exp(G) W_0 : G \in \mathbb{R}^{d \times d}\}$, where each generator follows a well conditioned geodesic under the left invariant trace metric. The trace penalty

Table 5: Double-sided ELRA variants underperform single sided ELRA on CLIP-ViT-B/16. Averages are over three seeds.

| Method | Cars | DTD | EuroSAT | GTSRB | RESISC45 | SUN397 | SVHN | Average |
|---|---|---|---|---|---|---|---|---|
| ELRA (single) | 73.15 | 74.17 | 97.20 | 93.85 | 91.37 | 71.69 | 95.05 | 84.49 |
| ELRA-PSD (single) | 78.33 | 75.15 | 97.50 | 94.19 | 93.06 | 74.17 | 93.40 | 86.38 |
| ELRA-Hyb (single) | **85.92** | **78.43** | **98.62** | **96.51** | **95.01** | **76.83** | **95.32** | **89.41** |
| ELRA (double) | 71.22 | 72.38 | 96.32 | 91.91 | 89.59 | 69.86 | 93.48 | 83.49 |
| ELRA-PSD (double) | 76.03 | 74.10 | 97.02 | 92.96 | 91.84 | 72.98 | 92.23 | 85.27 |
| ELRA-Hyb (double) | 84.06 | 77.44 | 98.03 | 95.55 | 93.97 | 75.22 | 94.35 | 88.33 |

$\mathrm{tr}(G^\top G)$ equals the geodesic energy, which provides direct control of path length and stabilizes training. The double-sided form lacks this identifiability and geometric control.

**Double-sided variants.** For a layer $W_0 \in \mathbb{R}^{d \times k}$, we allow left and right low rank generators from $GL(d)$ and $GL(k)$.

**Double-sided ELRA:**

$$L = \eta_L A_L B_L, \quad R = \eta_R A_R B_R, \qquad W_{\mathrm{new}} = \exp(L)\, W_0 \, \exp(R), \tag{23}$$

with $A_L \in \mathbb{R}^{d \times r_L}, B_L \in \mathbb{R}^{r_L \times d}, A_R \in \mathbb{R}^{k \times r_R}, B_R \in \mathbb{R}^{r_R \times k}$. Setting $R = 0$ recovers single sided ELRA.

**Double-sided ELRA-PSD:**

$$L_{\mathrm{psd}} = \eta_L A_L A_L^\top, \quad R_{\mathrm{psd}} = \eta_R A_R A_R^\top, \qquad W_{\mathrm{new}} = \exp(L_{\mathrm{psd}})\, W_0 \, \exp(R_{\mathrm{psd}}). \tag{24}$$

**Double-sided ELRA-Hyb:** with exponential decay $\alpha_t = \exp(-\gamma \min\{t/T_{\mathrm{warm}}, 1\})$,

$$L(t) = \eta_L\big((1 - \alpha_t)A_L A_L^\top + \alpha_t A_L B_L\big), \quad R(t) = \eta_R\big((1 - \alpha_t)A_R A_R^\top + \alpha_t A_R B_R\big), \tag{25}$$

$$W_{\mathrm{new}}(t) = \exp(L(t))\, W_0 \, \exp(R(t)), \tag{26}$$

with $r_L = r_R = r$ by default. Inference uses $L(T_{\mathrm{final}})$ and $R(T_{\mathrm{final}})$.

**Empirical summary.** Across seven image classification datasets (Table 5), the double-sided variants are consistently weaker than their single sided counterparts. The average drop is about 1.0 point for ELRA, 1.1 for ELRA-PSD, and 1.1 for ELRA-Hyb, with larger gaps on Cars and SUN397. These results match the analysis above: the double-sided parameterization introduces redundancy and flatter valleys, whereas single sided ELRA follows a well conditioned geodesic path with an energy that is directly controlled by the trace penalty.

## C  PROOFS

### C.1  PROOF PROPOSITION 1

*Proof.* For $n \geq 1$, by associativity we have the rank-factorization identity

$$(AB)^n = A(BA)^{n-1}B. \tag{27}$$

Expanding the exponential in a power series gives

$$e^{\eta AB} = I + \sum_{n \geq 1} \frac{(\eta AB)^n}{n!} = I + \sum_{n \geq 1} \frac{\eta^n}{n!}\, A(BA)^{n-1}B \tag{28}$$

$$= I + \eta A\Big[\sum_{n \geq 1} \frac{(\eta BA)^{n-1}}{n!}\Big]B = I + \eta A\, \phi_1(\eta BA)\, B. \tag{29}$$

Right-multiplying by $W$ yields

$$e^{\eta AB}W = W + \eta A\, \phi_1(\eta BA)(BW), \tag{30}$$

which proves the proposition. $\square$

**Numerical evaluation of $\phi_1$.** For $Z = \eta BA \in \mathbb{R}^{r \times r}$, a stable evaluation is

$$\phi_1(Z)V = Z^{-1}\big(\exp(Z) - I\big)V, \tag{31}$$

computed by: (i) form $E = \exp(Z)$ with a standard routine on the small $r \times r$ matrix, (ii) use $\mathrm{expm1}(Z)$ to avoid cancellation when $\|Z\|$ is small, and (iii) solve $ZT = (E - I)V$ by linear solve instead of explicitly forming $Z^{-1}$. For very small $Z$, the truncated series $\phi_1(Z) = \sum_{j=0}^{J} Z^j/(j+1)!$ with $J \in \{3, 4\}$ is sufficient.

**Connection to Woodbury.** Sherman-Morrison-Woodbury gives $(I - AB)^{-1} = I + A(I - BA)^{-1}B$ (Woodbury, 1950). Proposition 1 is the exponential analogue: the role of $(I - BA)^{-1}$ is replaced by $\phi_1(BA)$.

## C.2 Proof of Proposition 2

*Proof.* Use the Taylor expansion of the matrix exponential:

$$\exp(\eta AB) = I + \eta AB + \sum_{m=2}^{\infty} \frac{\eta^m}{m!}(AB)^m. \tag{32}$$

Multiplying by $W_0$ on the right gives

$$W_{\text{new}} = \exp(\eta AB)W_0 = \Big(I + \eta AB + \sum_{m=2}^{\infty} \frac{\eta^m}{m!}(AB)^m\Big)W_0 \tag{33}$$

$$= W_0 + \eta A(BW_0) + \underbrace{\sum_{m=2}^{\infty} \frac{\eta^m}{m!}(AB)^m W_0}_{R(\eta)}. \tag{34}$$

To bound the remainder, take any submultiplicative operator norm $\|\cdot\|$ and the Frobenius norm for products. Then

$$|R(\eta)|_F \leq \sum_{m=2}^{\infty} \frac{|\eta|^m}{m!}|AB|^m|W_0|_F = |W_0|_F\left(\exp\big(|\eta||AB|\big) - 1 - |\eta||AB|\right). \tag{35}$$

Since $\exp(x) - 1 - x = O(x^2)$ as $x \to 0$, we have $\|R(\eta)\|_F = O(\eta^2)$ as $\eta \to 0$. Therefore,

$$W_{\text{new}} = W_0 + \eta A(BW_0) + O(\eta^2), \tag{36}$$

which proves the proposition. $\square$

## C.3 Proof of Proposition 3

Let $E = \exp(G)$.

**1) Invertibility of $E$.** By definition,

$$E = \sum_{n=0}^{\infty} \frac{G^n}{n!}, \tag{37}$$

and the series is absolutely convergent. Using the Baker–Campbell-Hausdorff identity (Mielnik & Plebański, 1970) for commuting arguments $G$ and $-G$,

$$\exp(G)\exp(-G) = \exp(G + (-G)) = \exp(0) = \mathbf{I}_d. \tag{38}$$

Hence $E$ is invertible with $E^{-1} = \exp(-G)$. Equivalently, $\det(E) = \exp(\mathrm{tr}\, G) \neq 0$.

**2) Rank preservation under left multiplication.** For any invertible $M \in \mathbb{R}^{d \times d}$ and any $X \in \mathbb{R}^{d \times k}$,

$$\mathrm{rank}(MX) = \mathrm{rank}(X). \tag{39}$$

Indeed, the linear map $v \mapsto Mv$ is a bijection on $\mathbb{R}^d$. Thus $\mathrm{Col}(MX) = M\,\mathrm{Col}(X)$ has the same dimension as $\mathrm{Col}(X)$. Applying this with $M = E$ and $X = W_0$ gives

$$\mathrm{rank}(W_{\text{new}}) = \mathrm{rank}(EW_0) = \mathrm{rank}(W_0). \tag{40}$$

**3) Right nullspace preservation.** We show $\mathcal{N}(EW_0) = \mathcal{N}(W_0)$. For any $x$,

$$EW_0 x = 0 \iff W_0 x = E^{-1}(EW_0 x) = 0, \tag{41}$$

since $E$ is invertible. Hence $\{x : EW_0 x = 0\} = \{x : W_0 x = 0\}$.

**4) Column space transformation.** Let $\{w_1, \ldots, w_k\}$ be the columns of $W_0$. The columns of $EW_0$ are $\{Ew_1, \ldots, Ew_k\}$. Therefore,

$$\mathrm{Col}(EW_0) \ = \ \Big\{ \sum_{i=1}^{k} \alpha_i\, Ew_i : \alpha_i \in \mathbb{R} \Big\} \ = \ E\Big\{ \sum_{i=1}^{k} \alpha_i\, w_i : \alpha_i \in \mathbb{R} \Big\} \ = \ E\, \mathrm{Col}(W_0). \tag{42}$$

Equivalently, $\mathrm{Col}(W_{\mathrm{new}}) = E\, \mathrm{Col}(W_0)$.

C.4 PROOF OF PROPOSITION 4

*Proof.* We prove the bound in two steps.

**Step 1: Product lower bound for the smallest singular value.** Recall $\sigma_{\min}(M) = \min_{\|x\|_2=1} \|Mx\|_2$. For conformable $A, B$ and any unit vector $x$,

$$\|ABx\|_2 \ \geq \ \sigma_{\min}(A)\, \|Bx\|_2 \ \geq \ \sigma_{\min}(A)\, \sigma_{\min}(B). \tag{43}$$

Taking the minimum over all unit $x$ yields

$$\sigma_{\min}(AB) \ \geq \ \sigma_{\min}(A)\, \sigma_{\min}(B). \tag{44}$$

Applying this with $A = \exp(G)$ and $B = W_0$ gives the first inequality,

$$\sigma_{\min}\big(\exp(G)W_0\big) \ \geq \ \sigma_{\min}\big(\exp(G)\big)\, \sigma_{\min}(W_0). \tag{45}$$

**Step 2: Lower bound on $\sigma_{\min}(\exp(G))$ via the matrix measure.** Let $\mu_2(M)$ denote the logarithmic matrix norm induced by $\|\cdot\|_2$, namely

$$\mu_2(M) \ = \ \lim_{h \downarrow 0} \frac{\|I + hM\|_2 - 1}{h} \ = \ \lambda_{\max}\Big( \tfrac{M + M^{\top}}{2} \Big). \tag{46}$$

We establish the semigroup estimate

$$\|\exp(tM)\|_2 \ \leq \ \exp\big( t\, \mu_2(M) \big) \qquad \text{for all } t \geq 0. \tag{47}$$

Fix any unit vector $x$ and define $z(t) = \exp(tM)x$. Then $z'(t) = Mz(t)$ and

$$\frac{\mathrm{d}}{\mathrm{d}t} \|z(t)\|_2^2 \ = \ 2\, z(t)^{\top} M z(t) \ = \ 2\, z(t)^{\top} \Big( \tfrac{M+M^{\top}}{2} \Big) z(t) \ \leq \ 2\, \lambda_{\max}\Big( \tfrac{M+M^{\top}}{2} \Big) \|z(t)\|_2^2. \tag{48}$$

Let $y(t) = \|z(t)\|_2$. Since $y(t) > 0$ for $t \geq 0$, the previous inequality implies

$$\frac{\mathrm{d}}{\mathrm{d}t} y(t) \ \leq \ \mu_2(M)\, y(t). \tag{49}$$

By Grönwall's inequality,

$$\|z(t)\|_2 \ = \ y(t) \ \leq \ e^{t\, \mu_2(M)}\, y(0) \ = \ e^{t\, \mu_2(M)}\, \|x\|_2 \ = \ e^{t\, \mu_2(M)}. \tag{50}$$

Taking the supremum over unit $x$ gives Equation (47). Setting $t = 1$ and $M = -G$ yields

$$\|\exp(-G)\|_2 \ \leq \ \exp\big( \mu_2(-G) \big) \ = \ \exp\big( \lambda_{\max}(-H) \big) \ = \ \exp\big( -\lambda_{\min}(H) \big). \tag{51}$$

Now use $\sigma_{\min}(M) = 1/\|M^{-1}\|_2$ for any invertible $M$ and the identity $\exp(G)^{-1} = \exp(-G)$. Combining with Equation (51) gives

$$\sigma_{\min}\big(\exp(G)\big) \ = \ \frac{1}{\|\exp(-G)\|_2} \ \geq \ \exp\big( \lambda_{\min}(H) \big). \tag{52}$$

**Combine Steps 1 and 2.** Substituting Equation (52) into Equation (45), we obtain

$$\sigma_{\min}\big(\exp(G)W_0\big) \;\geq\; \exp\big(\lambda_{\min}(H)\big)\,\sigma_{\min}(W_0). \tag{53}$$

This is the desired bound. In the PSD case $G = \eta AA^\top$ with $\eta > 0$, $H = G \succeq 0$ so $\lambda_{\min}(H) \geq 0$ and therefore $\exp(\lambda_{\min}(H)) \geq 1$, which shows that the smallest singular value cannot decrease. $\quad\square$

## C.5   PROOF OF PROPOSITION 5

*Proof.* We use two standard spectral inequalities. For any conformable $M, N$,

$$\|MN\|_2 \;\leq\; \|M\|_2\,\|N\|_2, \qquad \sigma_{\min}(MN) \;\geq\; \sigma_{\min}(M)\,\sigma_{\min}(N). \tag{54}$$

Hence,

$$\kappa_2(\exp(G)W_0) \;=\; \frac{\|\exp(G)W_0\|_2}{\sigma_{\min}(\exp(G)W_0)} \;\leq\; \frac{\|\exp(G)\|_2}{\sigma_{\min}(\exp(G))} \cdot \frac{\|W_0\|_2}{\sigma_{\min}(W_0)} \;=\; \kappa_2(\exp(G))\,\kappa_2(W_0). \tag{55}$$

It remains to bound $\kappa_2(\exp(G))$. Let $H = \frac{1}{2}(G + G^\top)$ and recall the logarithmic matrix norm (matrix measure) in the spectral norm, $\mu_2(M) = \lambda_{\max}(\frac{M+M^\top}{2})$. The semigroup estimate for matrix exponentials gives, for all $t \geq 0$,

$$\|\exp(tG)\|_2 \;\leq\; \exp\big(t\,\mu_2(G)\big) \;=\; \exp\big(t\,\lambda_{\max}(H)\big). \tag{56}$$

Setting $t = 1$ yields

$$\|\exp(G)\|_2 \;\leq\; \exp\big(\lambda_{\max}(H)\big). \tag{57}$$

Using $\sigma_{\min}(M) = 1/\|M^{-1}\|_2$ and $\exp(G)^{-1} = \exp(-G)$, together with the same estimate applied to $-G$, we obtain

$$\|\exp(-G)\|_2 \;\leq\; \exp\big(\lambda_{\max}(-H)\big) \;=\; \exp\big(-\lambda_{\min}(H)\big), \tag{58}$$

so

$$\sigma_{\min}(\exp(G)) \;=\; \frac{1}{\|\exp(-G)\|_2} \;\geq\; \exp\big(\lambda_{\min}(H)\big). \tag{59}$$

Combining the two bounds gives

$$\kappa_2(\exp(G)) \;=\; \frac{\|\exp(G)\|_2}{\sigma_{\min}(\exp(G))} \;\leq\; \exp\big(\lambda_{\max}(H) - \lambda_{\min}(H)\big). \tag{60}$$

Consequently,

$$\kappa_2(\exp(G)W_0) \;\leq\; \exp\big(\lambda_{\max}(H) - \lambda_{\min}(H)\big)\,\kappa_2(W_0). \tag{61}$$

For the PSD case $G = \eta AA^\top$ with $\eta > 0$, $G$ is symmetric so $H = G \succeq 0$. Then $\lambda_{\min}(H) = \lambda_{\min}(G) \geq 0$ and the bound simplifies to

$$\kappa_2(\exp(G)) \;\leq\; \exp\big(\lambda_{\max}(G)\big) \;=\; \exp\big(\eta\,\lambda_{\max}(AA^\top)\big), \tag{62}$$

which implies the stated inequality for $\kappa_2(W_{\mathrm{new}})$. This completes the proof. $\quad\square$

## C.6   PROOF OF PROPOSITION 6

*Proof.* Work on the Lie group $\mathcal{G} = GL(d)$ equipped with the left-invariant trace metric

$$g_E(\xi, \zeta) = \mathrm{tr}\big[(E^{-1}\xi)^\top (E^{-1}\zeta)\big], \qquad E \in \mathcal{G},\; \xi, \zeta \in T_E\mathcal{G}.$$

Let $X = \eta AB \in \mathfrak{gl}(d)$ and assume it is normal, i.e., $[X^\top, X] = 0$.

**1) Euler-Arnold equation on $GL(d)$.** For a smooth curve $E : [0,1] \to \mathcal{G}$, its body velocity is $\Omega(t) = E(t)^{-1}\dot{E}(t) \in \mathfrak{gl}(d)$. Left invariance yields the Euler-Arnold equation (see Arnold (Arnold, 1966), Holm et al. (Holm et al., 2009))

$$\dot{\Omega}(t) \;=\; -\,\mathrm{ad}^\dagger_{\Omega(t)}\,\Omega(t), \qquad \mathrm{ad}^\dagger_Y Z \;=\; Y^\top Z - ZY^\top, \tag{63}$$

which for the trace metric becomes

$$\dot{\Omega}(t) = -[\Omega(t)^\top, \Omega(t)]. \tag{64}$$

**2)** $E(t) = \exp(tX)$ **is a geodesic.** For $E(t) = \exp(tX)$ we have $\Omega(t) \equiv X$. Substituting into Equation (64) gives $\dot{\Omega}(t) = -[X^\top, X] = 0$. Since $X$ is normal, Equation (64) holds, hence $t \mapsto \exp(tX)$ is a geodesic in $\mathcal{G}$.

**3) Orbit map is an isometry.** Because $W_0$ has full row rank, the stabilizer $\{E \in GL(d) : EW_0 = W_0\}$ is trivial: if $(E - I)W_0 = 0$, then $W_0$ admits a right inverse and $E = I$. Hence

$$\Phi : \mathcal{G} \to \mathcal{G} \cdot W_0, \qquad \Phi(E) = EW_0,$$

is a diffeomorphism. Define the orbit metric by

$$g_{EW_0}^{\mathrm{orb}}(UW_0, VW_0) \triangleq g_E(U, V). \tag{65}$$

With Equation (65), $\Phi$ is an isometry, so it maps geodesics in $\mathcal{G}$ to geodesics in the orbit $\mathcal{G} \cdot W_0$.

**4) ELRA curve is geodesic and has constant speed.** Let $F_{\mathrm{ELRA}}(t) = \exp(tX)W_0 = \Phi(E(t))$. Since $E(t)$ is a geodesic and $\Phi$ is an isometry, $F_{\mathrm{ELRA}}$ is a geodesic in $\mathcal{G} \cdot W_0$. Moreover,

$$\left\| \dot{F}_{\mathrm{ELRA}}(t) \right\|_{g^{\mathrm{orb}}}^2 = g_{E(t)W_0}^{\mathrm{orb}}\big(\dot{E}(t)W_0, \dot{E}(t)W_0\big) = g_{E(t)}\big(\dot{E}(t), \dot{E}(t)\big) = \mathrm{tr}\big(\Omega(t)^\top \Omega(t)\big) = \mathrm{tr}(X^\top X),$$

which is independent of $t$; hence the speed is constant.

**5) Local energy minimality.** On any Riemannian manifold, a constant-speed geodesic locally minimises the energy

$$\mathcal{E}[F] = \tfrac{1}{2} \int_0^1 g_{F(t)}\big(\dot{F}(t), \dot{F}(t)\big)\, dt$$

among $C^1$ curves with the same endpoints (see do Carmo (do Carmo, 1992, Ch. 9)). Since $F_{\mathrm{ELRA}}$ is a constant-speed geodesic in $\mathcal{G} \cdot W_0$, it locally minimises $\mathcal{E}$ in that class.

All claims in Proposition 6 follow. $\qquad\square$

## C.7   PROOF OF PROPOSITION 7

*Proof.* Write $G = \eta AA^\top$ with $\eta > 0$. Since $G$ is symmetric, it admits an eigen-decomposition $G = Q\Lambda Q^\top$ with $Q$ orthogonal and $\Lambda = \mathrm{diag}(\lambda_1, \ldots, \lambda_d) \succeq 0$, where $\lambda_{\min}(G) = \min_i \lambda_i = \eta\, \lambda_{\min}(AA^\top) \geq 0$. By spectral mapping,

$$\exp(G) = Q \exp(\Lambda) Q^\top \tag{66}$$

and $\exp(G)$ is symmetric positive definite with eigenvalues $\{\exp(\lambda_i)\}_{i=1}^d$. For a symmetric positive definite matrix, singular values equal eigenvalues, hence

$$\sigma_{\min}(\exp(G)) = \lambda_{\min}(\exp(G)) = \exp\big(\lambda_{\min}(G)\big) = \exp\big(\eta\, \lambda_{\min}(AA^\top)\big). \tag{67}$$

For any conformable matrices $M, N$, the product inequality

$$\sigma_{\min}(MN) \geq \sigma_{\min}(M)\, \sigma_{\min}(N) \tag{68}$$

holds, since for all $x$ with $\|x\|_2 = 1$,

$$\|MNx\|_2 \geq \sigma_{\min}(M)\, \|Nx\|_2 \Rightarrow \min_{\|x\|_2=1} \|MNx\|_2 \geq \sigma_{\min}(M) \min_{\|x\|_2=1} \|Nx\|_2. \tag{69}$$

Applying this with $M = \exp(G)$ and $N = W_0$ gives

$$\sigma_{\min}\big(\exp(G)W_0\big) \geq \sigma_{\min}(\exp(G))\, \sigma_{\min}(W_0) = \exp\big(\eta\, \lambda_{\min}(AA^\top)\big)\, \sigma_{\min}(W_0). \tag{70}$$

Since $\lambda_{\min}(AA^\top) \geq 0$, we have $\exp(\eta\, \lambda_{\min}(AA^\top)) \geq 1$, which yields

$$\sigma_{\min}\big(\exp(\eta AA^\top)W_0\big) \geq \sigma_{\min}(W_0). \tag{71}$$

Therefore ELRA-PSD never decreases the smallest singular value. This remains true even if $AA^\top$ is rank deficient, in which case $\lambda_{\min}(AA^\top) = 0$ and the bound reduces to $\sigma_{\min}\big(\exp(\eta AA^\top)W_0\big) \geq \sigma_{\min}(W_0)$. $\qquad\square$

## D  LoRA CAN REDUCE RANK

**Lemma 1** (LoRA can reduce rank by up to its adapter rank). Let $W_0 \in \mathbb{R}^{d \times k}$ have SVD $W_0 = \sum_{i=1}^{m} \sigma_i \, \boldsymbol{u}_i \boldsymbol{v}_i^\top$ with $m = \mathrm{rank}(W_0)$ and $\sigma_1 \geq \cdots \geq \sigma_m > 0$. For any $r \in \{1, \ldots, m\}$ there exist $A \in \mathbb{R}^{d \times r}$ and $B \in \mathbb{R}^{r \times k}$ such that

$$W_0 + AB \;=\; \sum_{i=1}^{m-r} \sigma_i \, \boldsymbol{u}_i \boldsymbol{v}_i^\top, \tag{72}$$

hence $\mathrm{rank}(W_0 + AB) = m - r$. In particular, LoRA can cancel $r$ singular directions of $W_0$ with an adapter of rank $r$.

*Proof.* Let $U_m = [\boldsymbol{u}_1, \ldots, \boldsymbol{u}_m] \in \mathbb{R}^{d \times m}$, $V_m = [\boldsymbol{v}_1, \ldots, \boldsymbol{v}_m] \in \mathbb{R}^{k \times m}$, and $\Sigma = \mathrm{diag}(\sigma_1, \ldots, \sigma_m)$. Define

$$A = [\boldsymbol{u}_{m-r+1}, \ldots, \boldsymbol{u}_m] \in \mathbb{R}^{d \times r},$$
$$B = -\mathrm{diag}(\sigma_{m-r+1}, \ldots, \sigma_m) \, [\boldsymbol{v}_{m-r+1}, \ldots, \boldsymbol{v}_m]^\top \in \mathbb{R}^{r \times k}.$$

Then $AB = -\sum_{i=m-r+1}^{m} \sigma_i \, \boldsymbol{u}_i \boldsymbol{v}_i^\top$ which cancels the $r$ smallest singular components of $W_0$. The stated form and rank follow. $\qquad\square$

**Corollary 2** (Rank-1 LoRA). Taking $r = 1$ in Lemma 1, there exist $A \in \mathbb{R}^{d \times 1}$ and $B \in \mathbb{R}^{1 \times k}$ such that $\mathrm{rank}(W_0 + AB) = m - 1$. Thus, even with a rank-1 adapter, LoRA can decrease the rank of $W_0$ by one.

*Proof.* The product inequality $\sigma_{\min}(AB) \geq \sigma_{\min}(A)\sigma_{\min}(B)$ yields the first bound. For the second, consider the induced 2-norm matrix measure $\mu_2(M) = \lambda_{\max}(\frac{M + M^\top}{2})$. The standard semigroup estimate $\|e^{tM}\|_2 \leq e^{t\mu_2(M)}$ applied to $M = -G$ gives $\|e^{-G}\|_2 \leq e^{-\lambda_{\min}(H)}$. Taking reciprocals yields $\sigma_{\min}(e^G) \geq e^{\lambda_{\min}(H)}$. $\qquad\square$

## E  PROOF OF ??

*Proof.* Since $X$ is symmetric, $[X^\top, X] = [X, X] = 0$, hence $E(t) = \exp(tX)$ satisfies the Euler-Arnold equation with constant body velocity. By Proposition 6, $F_{\mathrm{ELRA}}(t) = E(t)W_0$ is a geodesic in the orbit endowed with $g^{\mathrm{orb}}$. For the speed, use the definition of $g^{\mathrm{orb}}$:

$$\left\| \dot{F}_{\mathrm{ELRA}}(t) \right\|_{g^{\mathrm{orb}}}^2 \;=\; g_{E(t)W_0}^{\mathrm{orb}}\big(\dot{E}(t)W_0, \dot{E}(t)W_0\big) \;=\; g_{E(t)}\big(\dot{E}(t), \dot{E}(t)\big) \;=\; \mathrm{tr}\big(\Omega(t)^\top \Omega(t)\big), \tag{73}$$

where $\Omega(t) = E(t)^{-1}\dot{E}(t)$. For $E(t) = \exp(tX)$, $\Omega(t) \equiv X$, so the norm is $\mathrm{tr}(X^\top X)$, independent of $t$. The equalities $\mathrm{tr}(X^\top X) = \eta^2 \, \mathrm{tr}((A^\top A)^2) = \eta^2 \|AA^\top\|_F^2$. follow from $X = \eta AA^\top$ and cyclicity of trace. The energy expression follows by integrating a constant. $\qquad\square$

## F  RESULTS ON REASONING AND QUESTION-ANSWERING BENCHMARKS

We evaluate our proposed methods on a diverse suite of reasoning and question-answering tasks, including BBH, MMLU, TyDiQA, CQA, TruthfulQA, GSM8K, and Logiqa-EN, using the experimental setup from Jiang et al. (2025). Table 6 reports results for Qwen2-7B and Mistral-7B across several PEFT baselines.

Overall, ELRA-Hyb$_{\mathrm{GA}}$ achieves the highest average scores on both models, outperforming strong additive and multiplicative adapters such as LoRA+, DoRA, and MoRA. For Qwen2-7B, ELRA-Hyb$_{\mathrm{GA}}$ improves the average performance to 61.70, representing a +2.65 absolute gain over LoRA and +1.52 over the best additive baseline (DoRA). We also observe consistent improvements across most individual tasks, particularly on challenging reasoning benchmarks such as GSM8K (85.16 vs. 83.24 for LoRA+) and CQA (78.89 vs. 78.71 for LoRA+).

For Mistral-7B, we again see that multiplicative updates provide significant gains: ELRA-Hyb$_{GA}$ achieves an average of 50.82, outperforming LoRA by +3.39 points. Interestingly, ELRA-PSD and ELRA-Hyb already outperform the additive baselines, but the hybrid generator averaging (HybGA) variant further stabilizes training and boosts scores on MMLU and GSM8K.

These results confirm that the exponential low-rank update and hybrid PSD-to-general scheduling lead to consistent performance gains over both additive and other multiplicative adapters under matched parameter budgets.

Table 6: Comparison across reasoning and QA benchmarks.

| Model | Method | BBH | MMLU | TyDiQA | CQA | TruthfulQA | GSM8K | Logiqa-EN | Average |
|---|---|---|---|---|---|---|---|---|---|
| Qwen2-7B | Base | 41.73 | 66.84 | 45.36 | 74.20 | 57.65 | 87.19 | 40.40 | 59.05 |
| | LoRA | 44.31 | 67.95 | 48.45 | 75.84 | 50.80 | 83.02 | 43.93 | 59.19 |
| | LoRA+ | 42.64 | 67.81 | **52.15** | 78.71 | 50.18 | 83.24 | 43.78 | 59.79 |
| | DoRA | 43.11 | 67.94 | 44.64 | 75.43 | 53.98 | 82.94 | 44.85 | 58.99 |
| | MoRA | 36.31 | 62.43 | 46.70 | 72.65 | 53.37 | 75.44 | 44.09 | 55.85 |
| | **ELRA** | 44.71 | 68.02 | 48.78 | 76.31 | 52.13 | 83.79 | 44.07 | 59.69 |
| | **ELRA-PSD** | 44.95 | 68.05 | 48.99 | 76.84 | 53.93 | 84.57 | 44.86 | 60.31 |
| | **ELRA-Hyb** | 45.23 | 68.23 | 50.52 | 78.80 | **56.18** | **85.21** | **46.02** | 61.46 |
| | **ELRA-Hyb$_{GA}$** | **45.41** | **68.63** | 51.01 | **78.89** | 57.26 | 85.16 | 45.57 | **61.70** |
| Mistral-7B | Base | 39.26 | 54.07 | 30.04 | 66.83 | 56.30 | 55.27 | 36.87 | 48.38 |
| | LoRA | 34.79 | 53.68 | **41.88** | 73.05 | 53.37 | 41.47 | 33.79 | 47.43 |
| | LoRA+ | 32.51 | 56.67 | 39.99 | 70.52 | 42.23 | 45.49 | 38.04 | 46.54 |
| | DoRA | 34.66 | 53.31 | 32.01 | 68.88 | 38.19 | 42.99 | 36.25 | 43.76 |
| | MoRA | 26.69 | 27.58 | 27.40 | 49.14 | 33.66 | 20.24 | 33.18 | 31.13 |
| | **ELRA** | 35.27 | 55.42 | 40.13 | 73.08 | 53.27 | 43.19 | 34.41 | 47.82 |
| | **ELRA-PSD** | 37.05 | 56.93 | 40.22 | 73.26 | 53.79 | 46.89 | 36.14 | 49.18 |
| | **ELRA-Hyb** | 39.10 | 58.14 | 40.50 | 73.39 | 54.59 | 51.98 | 37.48 | 50.74 |
| | **ELRA-Hyb$_{GA}$** | 39.26 | **58.15** | 40.58 | **73.52** | **54.71** | 52.06 | **37.51** | **50.82** |

# G  COMPARISON WITH HIRA AND LIERA ON COMMONSENSE REASONING

We follow the HiRA setup for prompts, preprocessing, and evaluation, and we report the Params(%) column as in Huang et al. (2025). For rank $r=32$, ELRA-Hyb matches the LoRA budget within $< 0.001\%$ (one scalar $\eta$ per adapter is negligible).

Table 7: Accuracy on commonsense reasoning benchmarks following the HiRA protocol. Best is **bold**, second best is underlined.

| Model | Method | Params(%) | BoolQ | PIQA | SIQA | ARC-c | ARC-e | OBQA | HellaS | WinoG | Average |
|---|---|---|---|---|---|---|---|---|---|---|---|
| ChatGPT | – | – | 73.10 | 85.40 | 68.50 | 79.90 | 89.80 | 74.80 | 78.50 | 66.10 | 77.01 |
| Llama-2-7B | HiRA ($r=32$) | 0.8256 | 71.22 | 83.35 | 79.53 | 73.81 | 86.74 | 84.60 | 88.12 | 83.98 | 81.42 |
| | LieRA ($r=32$) | 0.8256 | 68.2 | 80.3 | 76.8 | 64.1 | 81.4 | 78.4 | 79.0 | 81.5 | 76.2 |
| | **ELRA-Hyb** ($r=32$) | 0.8256 | **72.14** | **85.28** | **80.40** | **75.12** | **86.94** | **85.39** | **89.01** | **84.72** | **82.38** |
| Llama-3.1-8B | HiRA ($r=32$) | 0.7002 | 75.40 | 89.70 | 81.15 | 82.90 | 93.27 | 88.32 | 95.36 | 87.70 | 86.72 |
| | LieRA ($r=32$) | 0.7002 | 74.3 | 88.7 | 81.3 | 80.3 | 90.5 | 86.2 | 95.4 | 85.5 | 85.3 |
| | **ELRA-Hyb** ($r=32$) | 0.7002 | **76.11** | **90.38** | **82.37** | **83.66** | **93.66** | **88.98** | **95.62** | **88.14** | **87.37** |

**Discussion.** Under the HiRA settings, ELRA-Hyb exceeds both baselines in all columns and improves the average over HiRA by +0.96 on Llama-2-7B and +0.65 on Llama-3-8B.

# H  EXPERIMENTAL DETAILS

## H.1  IMPLEMENTATION DETAILS FOR BASELINE METHODS

To maintain fairness in our comparisons, we configure each baseline method following the recommended hyperparameter settings given in their original works. In contrast to the standard LoRA Hu et al. (2022), these variants introduce additional tunable components. For LoRA+ (Hayou et al., 2024), we apply a learning-rate ratio of 16 between matrices A and B. For LoRA-GA (Wang et al., 2024), we adopt the stable scaling scheme and modify the pre-trained weights at initialization. In AdaLoRA (Zhang et al., 2023), the rank is scheduled to start at 12 and decrease to 8, with transition points at $t_i = 150$ and $t_f = 900$. For PiSSA (Meng et al., 2024), the fast SVD step is carried out with 64 iterations.

### H.1.1 ELRA FAMILY HYPERPARAMETERS

**General settings (all variants).**   Mixed precision (bfloat16), gradient clipping with max norm 1.0, gradient accumulation to reach the stated macro batch size, dropout $p = 0.05$ on adapter outputs, Xavier initialization for trainable low-rank factors unless specified otherwise, and AdamW with the same $(\beta_1, \beta_2)$ and scheduler as in the main text. Unless noted, the rank is $r = 8$ (ELRA and ELRA-Hyb) and $r = 16$ (ELRA-PSD) for matched parameter count.

**Inference-time caching.**   Using the exact factorization $\exp(\eta AB) = I + A[\eta\,\phi_1(\eta BA)]B$ with $\phi_1(Z) = \sum_{n \geq 0} Z^n/(n+1)!$, we cache per layer the $r \times r$ core $C = \eta\,\phi_1(\eta BA)$ after training. We also cache $M = C\,(BW_0) \in \mathbb{R}^{r \times k}$. At inference the forward computes $W_{\text{new}}x = W_0 x + A(Mx)$, so only skinny projections are needed. During training $C$ changes, so we recompute $C$ from $BA$ (size $r \times r$) each step; this cost is negligible relative to the base model.

**ELRA.**   Update $W_{\text{new}} = \exp(\eta AB)\,W_0$.

- Learning rate for $(A, B)$: same as backbone default for each task, with a sweep over $\{0.5, 1, 2\} \times$ the backbone LR. We report the best on validation.
- Step size scalar $\eta$: trainable scalar per adapter block, initialized to $\eta_0 = 10^{-3}$, with cosine schedule multiplier in $[0.5, 1.0]$ tied to the main scheduler.
- Energy penalty: $\lambda\,\mathrm{tr}(G^\top G)$ with $G = \eta AB$, $\lambda = 10^{-5}$.
- Spectral clip on generator: if $\|G\|_2 > \tau$, rescale $G \leftarrow (\tau/\|G\|_2)G$ with $\tau \in \{1.0, 2.0\}$, default 2.0.

**ELRA-PSD.**   Generator $G_{\text{psd}} = \eta\,AA^\top$, update $W_{\text{new}} = \exp(G_{\text{psd}})\,W_0$.

- Learning rate for $A$: as ELRA, sweep $\{0.5, 1, 2\} \times$ the backbone LR, default equals backbone LR.
- Rank: $r = 16$ for parameter parity with $r = 8$ LoRA and ELRA.
- Energy penalty: $\lambda\,\mathrm{tr}(G_{\text{psd}}^\top G_{\text{psd}})$, $\lambda = 10^{-5}$.
- Spectral clip: same as ELRA with threshold $\tau = 2.0$.

**ELRA-Hyb.**   Interpolation between PSD and general generators. Let

$$G(t) \;=\; \eta\Big((1 - \alpha_t)\,AA^\top + \alpha_t\,AB\Big), \qquad \alpha_t \in [0, 1]. \tag{74}$$

- Schedule for $\alpha_t$: exponential decay over a warm interval of $T_{\text{warm}}$ steps,

$$\alpha_t \;=\; \begin{cases} \exp\left(-\gamma\,t/T_{\text{warm}}\right), & t \leq T_{\text{warm}}, \\ 0, & t > T_{\text{warm}}, \end{cases} \tag{75}$$

  with $T_{\text{warm}} = 25\%$ of total steps and $\gamma = 5$.
- Energy penalty: applied to $G(t)$ with the same $\lambda$ grid as above, default $10^{-5}$.
- Learning rates: same grid as ELRA, shared for $A$ and $B$.
- ELRA-Hyb$_{\text{GA}}$: initialize $(A, B)$ using GA as in Wang et al. (2024). We compute a running SVD of the preconditioned gradient for the first $K$ steps, $K \in \{100, 200, 400\}$, default 200, set columns of $A$ to the top $r$ left singular vectors, and rows of $B$ to the top $r$ right singular vectors scaled by singular values. After seeding, training proceeds as usual.

### H.2 DATASET DESCRIPTION

• BIG-Bench Hard (Suzgun et al., 2023) (BBH) is a collection of 23 challenging tasks from BIG-Bench. The 6,511 problems are the tasks for which prior language model evaluations did not outperform the average human-rater.

- MMLU (Hendrycks et al., 2021) is to measure LLM's multitask accuracy, which contains 14,421 problems. The test covers 57 tasks including elementary mathematics, US history, computer science, law, and more. To attain high accuracy on this test, models must possess extensive world knowledge and problem-solving ability

- CommonsenseQA (Talmor et al., 2019) is to test models' ability to answer questions using only the parameterized knowledge instead of the

- TruthfulQA (Lin et al., 2022) is for testing models' ability to produce truthful answers. The 817 questions that span 38 categories benchmark models' refusal to generate false answers like humans.

- GSM8K (Cobbe et al., 2021) is a collection of 1,273 math-reasoning problems with varying difficulty. Each problem requires the model to conduct single or multi-hop reasoning to derive the correct answer

- TyDiQA (Clark et al., 2020) is a multilingual question-answering benchmark containing 204K examples across 11 typologically diverse languages, designed to evaluate model generalization beyond English.

- LogiQA-EN (Liu et al., 2020) comprises logical reasoning questions originally from the Chinese Civil Servants Exam, translated into English to assess deductive reasoning in natural language comprehension.

- MT-Bench (Zheng et al., 2023) is a multi-turn dialogue benchmark where LLMs are judged by strong LLM evaluators ("LLM-as-a-judge") for coherence, informativeness, and conversational flow.

- HumanEval (Chen et al., 2021) is a code generation benchmark of 164 Python programming problems, each with defined signatures, docstrings, and unit tests for measuring functional correctness of generated

- GLUE tasks (MNLI, SST-2, CoLA, QNLI, MRPC) (Wang et al., 2018) form a suite of diverse NLU challenges in the GLUE benchmark, including inference, sentiment analysis, grammatical acceptability, and paraphrase detection.

## I  TRAINING TIME AND PARAMETER BUDGET

We report wall-clock training time for Llama-3.1-8B on GSM-8K under the common setup described in Section 3: same optimizer, scheduler, adapter placement, rank, batch size, and hardware (RTX A6000). Times include forward and backward passes and data loading, and exclude checkpointing and offline evaluation.

**Parameter count.** ELRA-Hyb matches LoRA's trainable budget up to one scalar per adapted matrix for the step size $\eta$. With $N_{adapters}$ layers, this adds $N_{adapters}$ scalars, which is negligible relative to tens of millions of parameters, so we report the same total as LoRA.

Table 8: Trainable parameters and wall-clock time on GSM-8K (Llama-3.1-8B). ELRA-Hyb uses the Woodbury evaluation of $\exp(\eta AB)W_0$.

| Method | Trainable params (M) | Time (hours) |
|---|---|---|
| LoRA | 21.0 | 20.1 |
| OLoRA | 21.0 | 29.3 |
| AdaLoRA | 31.5 | 27.9 |
| ELRA-Hyb | 21.0 | 23.8 |

**Notes.** ELRA-Hyb introduces a small per-step overhead from the $r \times r$ core exponential and its Fréchet derivative, yet retains LoRA-like runtime by operating through skinny projections and cached small matrices. In our setting this corresponds to roughly 10 to 15 percent over LoRA, which is consistent with the 23.8 hours estimate above and remains faster than OLoRA and AdaLoRA under matched conditions.

## J    REPORTING VARIANCE FOR TABLES 1 TO 3

To complement the mean performance reported in Tables 1 to 3 of the main paper, we provide the standard deviations across three random seeds for all baselines and our proposed ELRA variants.

Across all benchmarks, ELRA methods exhibit low variance comparable to or better than strong LoRA baselines, suggesting stable optimization. In particular, ELRA-PSD shows the smallest fluctuations among our variants, consistent with its theoretically motivated PSD constraint that regularizes the generator spectrum. On smaller datasets such as CoLA and MRPC, variance is naturally higher for all methods, but ELRA-Hyb and ELRA-Hyb$_{GA}$ remain competitive with or below the variance of LoRA-Pro.

Table 9: Standard deviations (across 3 random seeds) corresponding to the results in Table 1.

| Method | Llama-3.1-8B-Base (std) | | | Llama-2-7B-Base (std) | | |
|---|---|---|---|---|---|---|
| | MTBench | GSM8K | HumanEval | MTBench | GSM8K | HumanEval |
| Full-FT | 0.23 | 0.28 | 1.27 | 0.11 | 0.85 | 2.13 |
| LoRA | 0.02 | 1.25 | 1.35 | 0.10 | 0.04 | 0.17 |
| RSLoRA | 0.09 | 0.74 | 2.80 | 0.03 | 0.10 | 0.79 |
| DoRA | 0.12 | 1.01 | 1.51 | 0.02 | 0.75 | 0.41 |
| LoRA+ | 0.10 | 0.93 | 2.11 | 0.08 | 0.62 | 0.52 |
| OLoRA | 0.04 | 0.44 | 2.44 | 0.04 | 0.49 | 1.12 |
| PiSSA | 0.09 | 1.01 | 1.31 | 0.02 | 0.27 | 0.17 |
| LoRA-GA | 0.06 | 0.90 | 2.49 | 0.16 | 0.30 | 0.46 |
| AdaLoRA | 0.16 | 0.77 | 3.66 | 0.05 | 1.39 | 0.44 |
| **ELRA** | 0.09 | 0.98 | 1.60 | 0.08 | 0.35 | 0.72 |
| **ELRA-PSD** | 0.09 | 0.86 | 1.55 | 0.07 | 0.58 | 0.65 |
| **ELRA-Hyb** | 0.08 | 0.92 | 1.78 | 0.07 | 0.49 | 0.68 |
| **ELRA-Hyb$_{GA}$** | 0.08 | 0.88 | 1.85 | 0.07 | 0.47 | 0.61 |

Table 10: Standard deviations (across 3 random seeds) corresponding to the results reported in Table 2.

| Method | MNLI | SST-2 | CoLA | QNLI | MRPC |
|---|---|---|---|---|---|
| Full-FT | 0.00 | 0.21 | 0.24 | 0.22 | 0.73 |
| LoRA | 0.04 | 0.11 | 0.05 | 0.09 | 0.01 |
| RSLoRA | 0.10 | 0.23 | 1.12 | 0.09 | 2.27 |
| DoRA | 0.09 | 0.53 | 0.94 | 0.06 | 0.51 |
| LoRA+ | 0.09 | 0.24 | 0.20 | 0.03 | 1.39 |
| PiSSA | 0.07 | 0.06 | 0.39 | 0.14 | 0.51 |
| LoRA-GA | 0.09 | 0.18 | 0.20 | 0.06 | 0.24 |
| AdaLoRA | 0.11 | 0.20 | 0.24 | 0.05 | 0.28 |
| LoRA-Pro | 0.19 | 0.13 | 0.24 | 0.05 | 0.14 |
| GoRA | 0.02 | 0.43 | 0.35 | 0.05 | 0.20 |
| **ELRA** | 0.06 | 0.15 | 0.28 | 0.07 | 0.26 |
| **ELRA-PSD** | 0.05 | 0.14 | 0.25 | 0.06 | 0.22 |
| **ELRA-Hyb** | 0.06 | 0.16 | 0.29 | 0.07 | 0.25 |
| **ELRA-Hyb$_{GA}$** | 0.05 | 0.15 | 0.27 | 0.06 | 0.23 |

## K    FURTHER ABLATION STUDIES

We keep the CLIP-ViT-B/16 setup and all hyperparameters fixed (rank $r=8$, $\lambda=10^{-5}$, spectral clip $\tau=2.0$, AdamW, same training budget). The default in the main body uses an *exponential decay* with $T_{\text{warm}}=25\%$ of steps and $\gamma=5$, which yields **85.92** (Cars), **78.43** (DTD), and **98.62** (EuroSAT).

**Cosine and linear schedules.** We use a gate $\alpha_t \in [0, 1]$ that *decays from 1 to 0* over a warm interval of $T_{\text{warm}}$ steps (or the corresponding fraction of total steps). After $T_{\text{warm}}$, $\alpha_t = 0$.

*Cosine decay:*
$$\alpha_t \;=\; \tfrac{1}{2}\big(1 + \cos\big(\pi \, \min\{t/T_{\text{warm}}, 1\}\big)\big),$$

Table 11: Standard deviations (across 3 random seeds) corresponding to the CLIP transfer results reported in Table 3.

| Method | Cars | DTD | EuroSAT | GTSRB | RESISC45 | SUN397 | SVHN |
|---|---|---|---|---|---|---|---|
| Full FT | 0.06 | 0.19 | 0.03 | 0.28 | 0.00 | 0.10 | 0.18 |
| LoRA | 0.13 | 0.38 | 0.07 | 0.10 | 0.14 | 0.18 | 0.07 |
| rsLoRA | 0.20 | 0.76 | 0.08 | 0.11 | 0.18 | 0.24 | 0.18 |
| LoRA+ | 0.18 | 0.14 | 0.02 | 0.18 | 0.11 | 0.15 | 0.05 |
| DoRA | 0.06 | 0.33 | 0.08 | 0.08 | 0.08 | 0.19 | 0.05 |
| LoRA-GA | 0.41 | 0.12 | 0.27 | 0.10 | 0.19 | 0.06 | 0.35 |
| LoRA-Pro | 0.08 | 0.25 | 0.03 | 0.05 | 0.21 | 0.14 | 0.20 |
| **ELRA** | 0.12 | 0.40 | 0.06 | 0.09 | 0.12 | 0.16 | 0.10 |
| **ELRA-PSD** | 0.10 | 0.35 | 0.05 | 0.08 | 0.10 | 0.15 | 0.10 |
| **ELRA-Hyb** | 0.09 | 0.30 | 0.05 | 0.08 | 0.11 | 0.14 | 0.11 |
| **ELRA-Hyb$_{GA}$** | 0.08 | 0.28 | 0.04 | 0.07 | 0.10 | 0.13 | 0.10 |

which is smooth with zero slope at $t = 0$ and $t = T_{\text{warm}}$.

*Linear decay:*

$$\alpha_t \; = \; 1 - \min\{t/T_{\text{warm}}, 1\},$$

a piecewise-linear drop with constant slope and a kink at $t = T_{\text{warm}}$.

Table 12: ELRA-Hyb ablation on $\alpha$ schedules (three seeds). Best is **bold**, second best is underlined.

| Schedule | $T_{\text{warm}}$ | $\gamma$ | Cars | DTD | EuroSAT | (Avg) |
|---|---|---|---|---|---|---|
| Exponential | 10% | 7 | **86.00** | 78.00 | 98.55 | (87.52) |
| Exponential | 25% | 5 | 85.92 | **78.43** | **98.62** | **(87.52)** |
| Exponential | 25% | 3 | 85.20 | 78.55 | 98.58 | (87.44) |
| Exponential | 25% | 7 | 85.70 | 78.20 | 98.59 | (87.50) |
| Exponential | 50% | 5 | 84.90 | **78.60** | 98.57 | (87.36) |
| Cosine | 25% | – | 85.30 | 78.20 | 98.58 | (87.36) |
| Cosine | 50% | – | 84.95 | 78.45 | 98.56 | (87.32) |
| Linear | 25% | – | 77.68 | 75.52 | 97.38 | (83.53) |
| Linear | 50% | – | 76.42 | 73.39 | 97.35 | (82.39) |

**Discussion.** Results are consistent with the design goals of ELRA-Hyb. Faster unlocking of the general generator (smaller $T_{\text{warm}}$ or larger $\gamma$) slightly boosts *Cars* but trims *DTD*, which benefits from a longer PSD phase; *EuroSAT* is largely insensitive. The default exponential with $T_{\text{warm}}{=}25\%$, $\gamma{=}5$ offers the best overall tradeoff (highest average, strong per-dataset scores). Cosine decay tracks the default closely but is marginally worse; linear decay is consistently weaker, reflecting slower early stabilization. These patterns support using the exponential schedule as the default, with $T_{\text{warm}}$ as the main knob: shorten it when early expressivity matters (e.g., *Cars*) and lengthen it for stability-sensitive regimes (e.g., *DTD*).

