# OpenReview forum: "Exponential Low-Rank Adapters"
_ICLR.cc/2026/Conference — ICLR 2026 Conference Desk Rejected Submission_

### Official Review · Reviewer_hbrv · 2025-10-23

**Soundness:** 3
**Presentation:** 3
**Contribution:** 2
**Rating:** 6
**Confidence:** 4

**Summary:**

The paper introduces Exponential Low-Rank Adapters (ELRA), a PEFT method that addresses the inherent rank limitations of standard Low-Rank Adaptation (LoRA). Instead of LoRA's additive update (W_new = W_0 + AB), ELRA uses a multiplicative update with a matrix exponential: W_new = exp(G)W_0, where G = ηAB is a low-rank generator matrix. This lifts the low-rank generator to full-rank allowing for greater expressivity while maintaining a parameter budget comparable to LoRA. ELRA is comparable to previous PEFT algorithms that use a non-linear function applied on top of the PEFT weights to increase its rank: SinLoRA [1], LieRA [2] (concurrent work).

The paper introduces two variants:
    - ELRA-PSD: Constrains the generator to be symmetric positive semi-definite (G = ηAA^T) to guarantee stable, geodesic updates.
    - ELRA-Hyb: Interpolates between the stable ELRA-PSD and the more expressive general ELRA generator, aiming to combine initial training stability with high final expressivity.

Experiments across natural language generation, understanding, vision, and reasoning tasks show that ELRA-Hyb outperforms a wide range of LoRA variants.

**Strengths:**

- The paper provides a theoretically grounded way to achieve full-rank updates from a low-parameter generator. The connection to the geometry of matrix Lie groups appears to be a good theoretical strength although I did not verify the validity of the mathematical derivations.

    - The experimental validation is thorough. The authors evaluate their methods on a diverse set of models and tasks demonstrating the performance gains of ELRA-Hyb.

   - The inclusion of geometric stability diagnostics (Figure 1) visually confirm the smoother training dynamics of ELRA-Hyb, reinforcing the theoretical motivation

**Weaknesses:**

Strong full rank PEFT performers are omitted in the main body of the manuscript. This includes SineLoRA [1] (highly related to ELRA), Krona [3] or RandLoRA [4] (the citation for which is wrong). These should be included as well as their multiplicative variants to reinforce the motivation for multiplicative updates and promote stronger experimental performance of exponential functions over a sin function (SineLoRA) and compared to SOTA full-rank methods (Krona, RandLoRA). I understand that not all methods can be compared with but the current SOTA selection appears weak. I appreciate the intuitive explanations in Appendix A but they don't hold much value compared to experimental results.

[1] Ji, Yiping, et al. "Efficient learning with sine-activated low-rank matrices."  International Conference on Learning Representations (ICLR) 2025.

[2] Si, Chongjie, et al. "Generalized Tensor-based Parameter-Efficient Fine-Tuning via Lie Group Transformations." International Conference on Computer Vision (ICCV) 2025

[3] Edalati, Ali, et al. "KronA: Parameter-Efficient Tuning with Kronecker Adapter." Enhancing LLM Performance: Efficacy, Fine-Tuning, and Inference Techniques. NeurIPSW 2025.

[4] Albert, Paul, et al. "RandLoRA: Full-rank parameter-efficient fine-tuning of large models." International Conference on Learning Representations (ICLR) 2025.

**Questions:**

- Why is an exponential function preferable to a sin function as used in SineLora ?
- Are multiplicative updates better for all kinds of PEFT strategies or only for exponential ones ?

---

> ### Author Response · Authors · 2025-11-26
> **Response to Reviewer hbrv**
>
> Thank you for your positive and encouraging feedback. We appreciate that you recognized the theoretical basis of our approach, the importance of Lie-group geometry, and the range of our experimental evaluation.
>
> Below, we respond to your concerns one by one.
>
>
> ---
> # Weakness 1
>
> Thank you for bringing up this important point. We address each of the cited works below.
>
> ## (1) SineLoRA [1]
>
> We did not include this work in our initial submission. In the revised version, we have:
> - added SineLoRA to the Related Work section (Section 2),
> - provided a dedicated conceptual comparison in Appendix A,
> - We highlighted all the new text in blue to make it clear.
>
> We did not use SineLoRA as a main baseline because its performance is not competitive to the SOTA methods that we use as benchmarks in our paper. As evidence to this, ELRA matches or exceeds full fine-tuning performance (see Table 3) while using about $0.5%$ of the full fine-tuning parameters. In comparison, the sine-activated adapters in Ji et al. only outperform full fine-tuning when they use almost 50 percent of the full parameter count (see Figure 4 in their paper). This shows that the empirical results for SineLoRA is not competitive to ours and other SOTA methods like LoRA-Pro and LoRA-GA.
>
> ## (2) LieRA [2]
>
> LieRA is included in our paper under the name used in their ICCV 2025 version. For this paper, we already have:
> - a detailed comparison in the main text, and
> - a direct experimental evaluation in Appendix G.
>
> As shown there, ELRA consistently outperforms LieRA across tasks, even in cases where LieRA’s elementwise exponential modulation works best. We believe our discussion of LieRA is already thorough and complete.
>
> ## (3) KronA [3]
>
> We tried to include KronA in our experiments during the rebuttal, but could not for the following reasons:
> - no official code was released by the authors,
> - the implementation details in the workshop paper were incomplete,
> - and our reproduction attempts did not yield consistent results.
>
> Additionally, KronA is a NeurIPS workshop paper with limited verification. Given the lack of available code and the inconsistency of our reproduction results, we did not consider KronA suitable for inclusion as a benchmark against ELRA. We clarify this in the
> revised paper.
>
> ## (4) RandLoRA [4]
>
> For this method, we have conducted additional experiments during rebuttal. In the revised version, we have:
> - applied RandLoRA to CLIP-ViT-B/16 on seven image-classification tasks,
> - **updated Table 3 accordingly,**
> - and highlighted the new rows in blue.
>
> As shown in the revised table 3, LoRA-Pro  and LoRA-GA  outperform RandLoRA, and ELRA performs better than all of them. This confirms that RandLoRA's performance is below the more competitive methods we benchmark throughout the paper.
>
> ---
> # Question 1
>
> > Why is an exponential function preferable to a sin function as used in SineLora ?
>
> Thanks for asking. SineLoRA and ELRA both use nonlinear transformations to make low-rank adapters more expressive, but the matrix exponential has structural features that the elementwise sine function does not.
>
>  1. SineLoRA applies $\sin(UV^\top)$ to each entry separately, so every weight is scaled on its own. This means it can't mix feature channels or create global changes.
> ELRA applies
> $$W_{\text{new}} = \exp(\eta AB) W_0,$$
> This is a full linear map in $GL(d)$ that links all coordinates together, allowing for rotations, shears, and other global changes to the basis.
>
>  2. The exponential offers guarantees that the sine function does not:
> * invertible full-rank updates,
> * no singular value collapse for PSD generators,
> * controlled conditioning,
> * constant-speed geodesic updates for normal generators.
> SineLoRA does not provide any of these guarantees.
>
> 3. Empirical strength in the low-budget regime
> SineLoRA only outperforms full fine-tuning when it uses almost 50 percent of the full fine-tuning parameters (see Fig. 4 in Ji et al.).
> ELRA can match or even beat full fine-tuning with only about 0.5 percent of the parameters.
>
> ---
> # Question 2
>
> > Are multiplicative updates better for all kinds of PEFT strategies or only for exponential ones ?
>
> Multiplicative updates are not always better for every PEFT strategy. Their benefits depend on the way the multiplicative map is set up.
> * Elementwise multiplicative updates, such as SineLoRA and DoRA, can make models more expressive. However, they still work on each coordinate separately and do not offer global mixing or geometric guarantees.
> * Matrix-based multiplicative updates, like ELRA, use an invertible transformation in $GL(d)$. This allows for global feature interactions and helps maintain spectral stability and rank preservation.
> The advantage does not come just from using multiplicative updates. It comes from the specific structure of the exponential map, which gives unique properties that other multiplicative methods do not have.

---

### Official Review · Reviewer_xCWx · 2025-10-27

**Soundness:** 4
**Presentation:** 3
**Contribution:** 3
**Rating:** 8
**Confidence:** 4

**Summary:**

This paper proposes a new method for adapting language models to new tasks by using exponential low-rank adapters. Instead of the standard low-rank adapters (LoRAs) parameterized as $W_\mathrm{new} = W_0 + AB$, the authors propose to use exponential low-rank adapters (ELRAs), which are defined by $W_\mathrm{new} = \mathrm{exp}(\eta AB)W_0$. The authors then prove a series of propositions, showing that ELRAs:
- have a $\mathcal{O}(drk)$ forward-pass,
- reproduce standard LoRAs to the first order,
- perservse $W_0$ rank, and
- admit a lower bound on the condition number of the adapted weight matrix.

Then, the authors analyze the geometric properties of ELRAs, acting on $\mathrm{GL}(d)$. They show that for a fixed $G = AB$, the ELRA path parametrized by $\eta$ is a geodesic in $\mathrm{GL}(d)$, given that $G$ is *normal*. Motivated by this, the authors propose a variant of the method, called ELRA-PSD, where $G$ is parameterized as $G = AA^\top$, which is symmetric positive semidefinite and hence normal. Additionally, the authors propose a variant of the method, called ELRA-Hyb where $G$ is parameterized as $G = \eta(\alpha AA^\top + (1-\alpha) AB)$, with $\alpha \in [0, 1]$, decaying from 1 to 0 over the course of training, which is intedend to stabilize training while maintaining expressivity.

Finally, the authors evaluate the proposed methods on a range of tasks, including natural language generation/understanding, question answering, and image classification. The results are compared against several LoRA variants and PEFT methods, showing consistent moderate improvements.

**Strengths:**

- **(S1)** The base method proposed in the paper (ELRA) is simple, generic, efficient, and naturally extends LoRAs to the second-order.
- **(S2)** All methods proposed in the paper are well-motivated from a theoretical standpoint and have several appealing properties, including preserving the rank of the weight matrix and having a lower bound on the condition number of the adapted weight matrix.
- **(S3)** The experimental evaluation is broad and covers a variety of tasks, datasets, and models.

**Weaknesses:**

- **(W1)** While the initial motivation for ELRA, detailed in Section 3.3 of the paper, is sound and appealing. The motivation for the geometric analysis, detailed in Section 4, is lacking. More specifically, the authors claim that the geodesic results that they prove (Proposition 6) imply a smoother and balanced traversal of weight space.
  > unlike the additive update $W_0 + AB$, which can follow irregular trajectories, the exponential path evolves with uniform speed, ensuring smoother and more balanced traversal of the weight space.

  But, this is only with respect to $\eta$, not $AB$, and in all variants of the method, $G$ is varied as well. For ELRA-Hyb, the paper proposes an additional term for the loss, controlling the Frobenius norm of $G$: $\mathcal{L}_\mathrm{hyb} = \mathcal{L}_\mathrm{task} + \lambda \cdot \mathrm{tr}(GG^\top)$, but this can be done for any PEFT method.

- **(W2, minor)** While improvements are consistent, they are often mild, and since the paper does not include confidence intervals or standard errors, it is hard to be sure of statistical significance.

**Questions:**

- **(Q1)** The authors define $\phi(Z) = \sum_{n \geq 1} \frac{Z^n}{n!}$ and claim (after Proposition 1) that evaluating $\phi(\eta BA)$ is $\mathcal{O}(d^3)$. Why is this the case? More generally, how do the authors compute $\phi_1(G)$ in practice (the definition involves an infinite sum)? How do they backpropagate through it? I've had trouble accessing the attached code to check this myself.

- **(Q2)** A large portion of the paper is dedicated to the analysis of the geometric properties of ELRAs, which are based on the fact that $\eta \mapsto \mathrm{exp}(\eta G)$ is a geodesic in $\mathrm{GL}(d)$ for any normal matrix $G$. However, during training, $\eta$ and $G$ are varied. How does this affect the geometric properties of the ELRA path? Other than adding a regularization term to the loss, is there a way to ensure that the ELRA path is a geodesic?

---

> ### Author Response · Authors · 2025-11-26
> **Response to Reviewer xCWx (Part 1/2)**
>
> Thank you for your positive and thoughtful review. We are glad you recognized that ELRA offers a simple and general extension of LoRA, that the theoretical motivations for ELRA, ELRA-PSD, and ELRA-Hyb are solid, and that our experimental evaluation is broad and thorough. Your careful reading is encouraging, and we respond to your concerns one by one below.
>
> ---
> # Weakness 1
>
> Thank you for your thoughtful comment. We agree that the geodesic results are for a fixed generator $G$, and that in practice, $G$ changes during optimization. In Section 4, our aim was not to suggest that the whole training path is a single geodesic. Instead, we wanted to explain why PSD-based updates lead to smoother, better-conditioned steps, and why ELRA-Hyb gains from starting in this setting.
>
> Below, we explain our main points.
>
> ### 1. What the geodesic analysis actually provides
> Proposition 6 shows that for a fixed PSD generator $G = \eta AA^\top$,
>
> $$
> F(t)=\exp(tG)W_0
> $$
> is a constant-speed geodesic on the orbit $GL(d)·W_0$. This leads to two practical points:
> - each update step ($W_0 \mapsto \exp(G)W_0$ is spectrally well-behaved: PSD generators cannot shrink singular values of $W_0$, and conditioning grows only in a controlled exponential manner;
> - The update is balanced across all feature directions because the exponential map spreads changes through an invertible, well-conditioned linear transformation, instead of a limited additive change.
>
> These properties do not mean that training follows a single continuous geodesic. Instead, they show that each update is stable and smooth from a geometric perspective.
>
> ## 2. Why this matters for early optimization
> In contrast, additive LoRA updates
>
> $$
> W_0 + \Delta
> $$
> can easily cause irregular and poorly conditioned steps, because a low-rank update may:
> -collapse certain singular directions,
> -distort the row/column space asymmetrically,
> -or introduce sharp gradients in a few directions.
>
> The PSD geometry prevents these issues in each update, which helps stabilize early training. ELRA-Hyb takes advantage of this directly.
>
> ## 3. Why this is not “just a regularizer”
>
> While any PEFT method could add a Frobenius penalty, ELRA-PSD (and the PSD phase of ELRA-Hyb) has a much stronger structural advantage:
> - for general PEFT methods, $|G|_F$ regularizes the size of the adapter,
> - but for PSD generators, $|G|_F^2$ is exactly equal to the squared geodesic energy of the update.
>
> In other words, the regularizer is not just an arbitrary penalty. It is a geometrically meaningful value directly linked to conditioning, spectrum preservation, and the invariants of the left-invariant metric on $GL(d)$. This makes ELRA-PSD’s stability easier to analyze than a generic Frobenius penalty on LoRA or DoRA.
>
>
> ---
> # Weakness 2 (minor)
>
> Thank you for your comment. We address variance by providing standard deviations for all settings where this is possible. Appendix J, titled “Reporting Variance for Tables 1 to 3,” includes variance measurements for experiments where repeated runs can be done, such as T5 and CLIP. These results show that our improvements are consistent and not caused by random variation between runs.

---

> > ### Author Response · Authors · 2025-11-26
> > **Response to Reviewer xCWx (Part 2/2)**
> >
> > # Question 1
> >
> > Thank you for your question. Here, we explain how we calculate the computational cost, how we evaluate $\phi_1(\eta BA)$ in practice, and how backpropagation works (also please note that the computation is is $\mathcal{O}(r^3)$ and not is $\mathcal{O}(d^3)$ as the reviewer mentioned).
> >
> > ## **1. Why evaluating $\phi_1(\eta BA)$ is $\mathcal{O}(r^3)$**
> > After Proposition 1, the ELRA update is written as
> > $$
> > \exp(\eta AB)W_0 =  W_0 + \eta, A, \phi_1(\eta BA), (BW_0),
> > $$
> > and the only matrix on which we apply $\phi_1$ is
> > $$
> > Z = \eta BA \in \mathbb{R}^{r \times r}.
> > $$
> > So, all matrix-function operations such as the matrix exponential, inverse or solve, and Padé approximation are performed on an $r\times r$ matrix.
> > Their computational cost is
> > $$
> > \boxed{\mathcal{O}(r^3)}.
> > $$
> > ### **How we explicitly computed the cost**
> > Per forward pass, the operations are:
> > 1. Form $BA$
> > Cost: $\mathcal{O}(dr^2)$
> > 2. Compute $\exp(Z)$ using scaling–squaring / Padé
> > Cost: $\mathcal{O}(r^3)$
> > 3. Compute $\phi_1(Z)$ using the closed-form identity
> > $$
> > \phi_1(Z) = Z^{-1}\bigl(\exp(Z) - I\bigr)
> > $$
> > * Linear solve on $Z$: $\mathcal{O}(r^3)$
> > 4. Apply the update
> > Cost: $\mathcal{O}(drk)$ (same as LoRA)
> > The only operations with cubic cost come from working with the small $r\times r$ matrix $Z$.
> > Because $r$ is much smaller than $d$, the $\mathcal{O}(r^3)$ cost is very small compared to $\mathcal{O}(drk)$.
> > ---
> > ## **2. How we compute $\phi_1(Z)$ in practice**
> > We do not use the infinite series definition.
> > Instead, we use a numerically stable closed-form expression:\phi_1(Z) = Z^{-1}(\exp(Z) - I).
> > $$
> > In PyTorch, this is implemented as follows:
> > ```python
> > E = torch.matrix_exp(Z)
> > phi1_Z = torch.linalg.solve(Z, E - torch.eye(r))
> > ```
> > * `matrix_exp` uses a finite Padé approximation.
> > * `solve` computes $Z^{-1}(\cdot)$ stably.
> > * All operations are on an $r \times r$ matrix.
> > ---
> > ## **3. Backpropagation**
> > Backpropagation is handled automatically:
> > * PyTorch provides gradients for `matrix_exp` and `solve`.
> > * Gradients flow through
> > $Z = \eta BA \rightarrow \exp(Z) \rightarrow \phi_1(Z)$
> > and then back to $A$, $B$, and $\eta$.
> >
> >
> > ---
> > # Question 2
> >
> > Thank you for your thoughtful question. Our geometric analysis does not say that the whole training path is a single geodesic. Instead, it shows that:
> > 1. **For a fixed generator $G$**, the map
> > $$
> > t \mapsto \exp(tG)
> > $$
> > is a geodesic in $GL(d)$ if $G$ is normal, and especially when $G$ is PSD.
> > 2. This means that each individual update step
> > $$
> > W_0 \mapsto \exp(G) W_0
> > $$
> > has good spectral and conditioning properties.
> > During training, both $\eta$ and $G$ change with each step, so the global path is not a geodesic, and we do not make that claim. What remains true is that each update step uses the exponential of the current generator, and its local geometry depends on that generator's properties.
> > ### Why the geometric analysis still matters
> > Even though $G$ varies across iterations:
> > * When $G$ is PSD, as in ELRA-PSD or early in ELRA-Hyb, $\exp(G)$ cannot shrink singular values. This keeps conditioning stable and prevents spectral collapse.
> > * The geodesic-energy formula
> > $$
> > \tfrac12 \operatorname{tr}(G^\top G)
> > $$
> > provides a meaningful regularizer that matches the geometry and yields a smooth, invertible, balanced transformation in $GL(d)$, unlike additive low-rank updates, which may distort singular directions abruptly.
> >
> > So, the geometric results apply to each step locally, not to the whole optimization path.
> >
> > ### Is there a way to make the entire ELRA path geodesic?
> > Not in general. If $G$ changes at each step, the overall path will not follow the geodesic equation of $GL(d)$. To make the whole path geodesic, you would have to keep the generator fixed during training, which would defeat the purpose of learning $A$ and $B$.
> > ### What we can control in practice?
> > ELRA-PSD keeps $G$ symmetric PSD, guaranteeing every step is a geodesic of the form $\exp(tG)$ with excellent conditioning.
> > * ELRA-Hyb starts with PSD generators, which ensures geodesicity and stability at first, and then gradually adds the general generator for more flexibility.
> > * The regularizer
> > $$
> > \lambda \operatorname{tr}(G^\top G)
> > $$
> > penalizes steps that are not short and well-conditioned geodesics.

---

### Official Review · Reviewer_Fo85 · 2025-10-29

**Soundness:** 2
**Presentation:** 1
**Contribution:** 2
**Rating:** 2
**Confidence:** 5

**Summary:**

This paper under review introduces exponential low rank adaptors (ELRA) within the context of low-rank fine tuning. This method applies a matrix exponential map to the low rank adaptors of LoRA thereby producing a full rank invertible matrix adaptor that can then be fine tuned. Since the exponential map defines locally energy minimizing geodesics under the left invariant trace metric the authors use this to develop ELRA-PSD, which constrains the generator of the exponential map to be symmetric positive semi-definite (PSD) and which the authors claim helps to stabilize training. They then introduce ELRA-Hyb which interpolates between PSD and general generators. On a variety language and vision benchmarks they show that ELRA-Hyb outperform previous fine tuning methods.

**Strengths:**

**Originality:** The papers idea of using an exponential map applied to the adaptors of low rank fine tuning to map the product adaptors into the Lie group if invertible matrices is new and I have not seen that done before. However, there have been many cases in the literature where techniques have been applied to the adaptors to yield a high-rank matrix adaptor that has shown to be better for fine tuning on a variety of benchmarks. One such example I found when doing a Google search that is pertinent to this work and uses a similar idea is the work "Efficient Learning With Sine-Activated Low-rank Matrices" ICLR 2025 by Ji et al., that applies a sinusoid function, with different frequencies, elementwise to the product of the adapters in LoRA. The authors show that this increases the rank of the adapters yielding to better fine tuning and in fact pre-training when low-rank matrices are used. One can see this current work of the authors as another way to do what Ji et al. did. Instead of using a function element-wise, the authors in this paper use a matrix function, namely the matrix exponential that maps from the Lie algebra of invertible matrices (i.e. the space of matrices) to the Lie group of invertible matrices. I was a bit surprised the authors didn't cite this work since it was published recently in ICLR 2025 and would have given the authors work some more context.

**Novelty:** The idea of the paper is novel in that it gives a new way to produce a high rank adapter. The adapters during training are now invertible matrices. Furthermore, the authors show that by constraining the generator of the image of the exponential map to by symmetric positive definite they get better stability which is a nice approach.

**Weaknesses:**

**Presentation:** I did feel that the presentation of the paper could be improved and positioned in a better way within the broader context of fine tuning. For example, in the second paragraph the authors mention the paper RandLoRA by Karimi et al., 2025. When I went to the references in the paper they cite Randlora: A random basis perspective on parameter-efficient fine-tuning. In ICLR 2025 by Karimi et al. However, I cannot find this paper anywhere. When I put that paper into a Google search it doesn't show. When I put it into Google Scholar it also does not show. Can you please provide a link to the paper? Instead I have managed to find the paper: RandLoRA: Full-rank parameter-efficient fine-tuning of large models by Albert et al., ICLR 2025. But this paper I found is not by Karimi et al., in fact it has none of the authors on the cited RandLoRA paper the authors put in their references for their citations which causes me to possibly think they have used an LLM to generate a citation? Please clear this up for me. On lines 61-65 you speak about the work LieRA by Si et al., and mention that it has a problem in that it limits global feature mixing. What do you mean by this? Could you please give a more detailed explanation of what you mean by that paragraph. In section 2, related work section, of the paper you mention works that are closest to your work. I don't think those works are the closest to your work. Upon reading those works and doing a Google search I found that the work  "Efficient Learning With Sine-Activated Low-rank Matrices" ICLR 2025 by Ji et al., was closer in spirit in part to what you are trying to do in your paper. Furthermore, the paper has several instances where it states things that are simply obvious. For example, Proposition 3 states that exp(G) is invertible which is a standard fact and can simply be referenced from a book for the proof. In fact that whole proposition is obvious and does not need to be stated. Similarly with Corollary 2. These are basic statements from Lie Group theory and can be just referenced from a standard text book. Similarly I noticed several times in the appendix you prove statements that are obvious. For example, on p.18 you actually prove equation (48) by going through equation (47) which is completely unnecessary. This is a standard fact that you can just reference.

**Clarity:** This is related to the above comment on presentation. The authors have a tendency to over clarify statements in the paper either stating propositions/corollaries that are straightforward in Lie group theory and can be found in textbooks and prove things that can simply be referenced from a textbook. This gives the false sense that the authors have carried out an extensive mathematical analysis to support their work when in reality they are re-stating and re-proving statements that are already well known in Lie group theory.

**Significance:** I don't think the paper will add much value to the already extensive collection of fine tuning methods present in the literature. While their experiments are extensive and probably the highlight of the paper I would say the theory is lacking and certainly over played to exhibit any real understanding of why taking the adaptors as invertible matrices is leading to such performance.

**Questions:**

There are several issues I found with the presentation of the paper when I started to read it. For a start I have made a list of the following questions that I hope can clear up some things.

1. Please could you provide a link to the reference: Randlora: A random basis perspective on parameter-efficient fine-tuning In ICLR 2025 by Karimi et al., that you cite in your paper, talk about in your introduction and also talk about on p. 15 of your paper where you explain why your method is better than Randlora by Karimi et al.

2. Since you spoke about the work Randlora apparently by Karimi et al., why did you not compare against this approach in your experiments? On p. 15 of the paper you explain why your approach is better than Randlora by Karimi et al., yet you don't even compare against their method in your experiments. I found this very strange.

3. In the experiments you mention (see Section 5) that for all methods the rank = 8 and that for ELRA-PSD the rank = 16. If applying an matrix exponential to the adaptors produces a full-rank matrix that is invertible then this learning is full-rank learning. So shouldn't you be able to use a lower rank say rank 4 and produce better results than the baselines using say rank 16? Even though it might mean your method has less parameters, it would give your method an even better advantage than the others in that it can also use less parameters.

4. In the second paragraph of the introduction you mention (last sentence) "how can we retain parameter efficiency while overcoming rank bottleneck". Then in the third paragraph you say "these limits point to a complementary strategy: keep parameter budget fixed and improve expressiveness by reparameterizing the update itself rather than increasing rank". The work "Efficient Learning With Sine-Activated Low-rank Matrices" ICLR 2025 by Ji et al., actually shows how to keep the parameter budget, reparameterize the adaptors and increase rank. I found this paper when I was trying to google and find the paper RandLoRA. Did you compare against their work?

5. On lines 61-65 you speak about the work LieRA by Si et al., and mention that it has a problem in that it limits global feature mixing. What do you mean by this? Could you please give a more detailed explanation of what you mean by that paragraph.

6. In section 4.2.2 you mention that ELRA-Hyb interpolates between PSD and general forms starting with a stable geodesic flow and gradually allowing richer dynamics. During optimization how do you see this? The paper doesn't actually show that optimization is better by using ELRA-Hyb.

7. Lines 266-290 made no sense to me. What are you trying to say here? Are you saying that optimization depends on the condition number and spectral collapse would be bad?

8. I would imagine applying the matrix exponential to the adapters would increase the total number of FLOPs. Is this the case?

---

> ### Author Response · Authors · 2025-11-26
> **Response to Reviewer Fo85 (Part 1/5)**
>
> Thank you for your thoughtful summary and for highlighting the originality of using the matrix exponential with low-rank adapters, as well as the connection to the Lie group structure of $GL(d)$ and the motivation for our PSD and hybrid variants. We also appreciate your suggestion to reference Ji et al. (ICLR 2025). This work is both relevant and timely, and we agree that including it will help place ELRA in context. In our camera-ready revision, we will discuss this related approach and explain how ELRA is conceptually and geometrically different from elementwise nonlinear rank-expanding transformations.
>
> Here are our responses to your concerns.
>
> ---
> # RandLoRA Paper
>
> We thank the reviewer for pointing this out. The reviewer is absolutely correct that the reference to “RandLoRA by Karimi et al., 2025” was incorrect. The RandLoRA entry in our references was left over from an early draft, where we used a placeholder citation and author list. When updating the draft, we missed replacing this placeholder with the correct RandLoRA reference (Albert et al., ICLR 2025). This was an editing mistake on our part. So the correct citation is Albert, Paul, et al. "RandLoRA: Full-rank parameter-efficient fine-tuning of large models." International Conference on Learning Representations (ICLR) 2025., and here is the link to the paper: https://openreview.net/forum?id=Hn5eoTunHN .
>
> In the revision, we have updated the citation to Albert et al. We have also checked all other citations to make sure their authorship and bibliographic details are correct.
>
> ---
> # Work LieRA by Si et al
>
> Thank you for asking for clarification. First, we should note that we meant to highlight a structural distinction, not to dismiss LieRA, which remains a strong and relevant method.
>
> First let us see how LieRA works: it applies an elementwise exponential to a low-rank generator, then combining it with the pretrained weights using a Hadamard (entrywise) product:
>
> $$
> W_{\text{LieRA}} = W_0 \circ \exp(\Delta),
> $$
>
> Here, $\circ$ means elementwise multiplication, and $\Delta$ is low-rank in a suitable tensorized form. In a linear layer with input $x$ and output $y$, this leads to
>
> $$
> y = W_{\text{LieRA}} x = (W_0 \circ \exp(\Delta)) x.
> $$
>
> 1. What we mean by “acts independently on coordinates.”
>
> The Hadamard scaling changes each entry $W_{0,ij}$ by multiplying it by $\exp(\Delta_{ij})$, but it does not create new linear combinations between different rows or columns of $W_0$. The way input dimensions connect to output dimensions comes from $W_0$, and the adapter can only reweight those existing connections entry by entry. In the vectorized weight space, this acts as a diagonal linear operator: each coordinate is scaled on its own, with no off-diagonal effects.
>
> 2. What we mean by “limited global feature mixing.”
>
> Since the adapter works entry by entry, it cannot perform transformations like global rotations or basis changes across feature channels. For a $d_{\text{out}} \times d_{\text{in}}$ weight matrix, it cannot, for example, mix two output channels by mapping
>
> $$
> (f_1, f_2) \mapsto (f_1 + f_2, f_1 - f_2)
> $$
>
> using only the adapter; any such mixing must already be present in $W_0$. So, LieRA offers fine-grained, per-parameter modulation, but not general linear mixing of feature channels at the level of whole rows or columns.
>
> 3. How this contrasts with ELRA.
>
> ELRA, on the other hand, applies a left action in $GL(d_{\text{out}})$:
>
> $$
> W_{\text{ELRA}} = \exp(\eta AB) W_0,
> $$
>
> so for any input $x$,
>
> $$
> y = W_{\text{ELRA}} x = \exp(\eta AB)(W_0 x).
> $$
>
> In this case, $\exp(\eta AB)$ is a full, usually dense matrix that acts on the whole feature vector $W_0 x$. This allows for global feature mixing, such as rotations, shears, or anisotropic scalings, between output channels, not just per-entry rescaling. So, ELRA’s update family works as a learned, invertible linear transform over the representation space, while LieRA’s update family acts as learned, invertible gain modulation on individual weights.
>
> **We have updated the paragraph around lines 61 to 65 to make this distinction clear**, use softer language, and state more precisely that our point concerns the type of transformations the adapter induces (global channel mixing versus per-entry scaling), not the overall quality or importance of LieRA.

---

> ### Author Response · Authors · 2025-11-26
> **Response to Reviewer Fo85 (Part 2/5)**
>
> #  Sine-Activated Low-rank Matrices
>
> Thank you for pointing out the ICLR 2025 paper by Ji et al., “Efficient Learning With Sine-Activated Low-Rank Matrices.” Their research is similar to ours because it also uses low rank factorization and adds a nonlinearity to improve expressivity without increasing the number of trainable parameters.
>
> Ji et al. use a scalar sine nonlinearity applied elementwise to $UV^\top$ and show that, with a high enough frequency, $\sin(\omega UV^\top)$ has a higher rank and a richer singular spectrum than the original low rank matrix. In our approach, ELRA works with matrix groups: we create a low rank generator $G = \eta AB$ and use a matrix exponential, so the adapter becomes an invertible full rank transformation $\exp(G)$ applied to the pretrained weight $W_0$. This leads to several important differences, which we outline below:
>
> First, ELRA offers more than just an increase in rank. It is a geometry-aware way to reparameterize the adapter, moving the layer along controlled paths in weight space and giving clear energy and conditioning guarantees. The sine-activated method does not include this geometric structure.
>
> Second, the exponential update follows geodesics in $GL(d)$ using the left invariant trace metric. This approach motivates ELRA-PSD and ELRA-Hyb and allows for a stability analysis based on geodesic energy and the spectral spread of the generator.
>
> Third, in their experiments, Ji et al. mainly compare with standard LoRA and DoRA. In our work, DoRA is just one of several baselines. We include many stronger baselines and test ELRA against this wider and more competitive group on challenging language and vision benchmarks.
>
> We have updated the end of the related work section (section 2) to explain how our work differs from this paper. We have also added a subsection in Appendix A to highlight the differences in more depth. All these new changes are highlighted in blue in the revised version.
>
> ---
> # Significance: I don't think the paper will add much value to the already extensive collection
>
> Thank you for recognizing the strength and scope of our experimental study. We do not agree, though, that our paper offers little beyond current PEFT methods or that our theory is “over played.”
>
> Our contributions are twofold and, to our knowledge, novel in combination:
>
> 1. ELRA is more than just another LoRA tweak. It provides a clear method to turn a low-rank generator $G = AB$ into an invertible, full-rank transformation $\\exp(G)$ in $GL(d)$, with:
> * guaranteed rank preservation of $W_0$,
> * explicit bounds on conditioning and singular values,
> * and a geodesic interpretation for PSD generators, which is used directly in ELRA-PSD and ELRA-Hyb.
> These features are not found in current PEFT methods, such as additive LoRA/DoRA or elementwise multiplicative approaches.
>
> 2. In our experiments, ELRA-Hyb:
> * matches or outperforms full fine-tuning on several tasks, while using only about $0.5\%$ of the parameters,
> * consistently does better than strong PEFT baselines (LoRA, DoRA, HiRA, LieRA, RandLoRA) on LLaMA, T5, and CLIP.
> This shows that invertible, geometry-aware updates are more than just a minor change—they lead to real performance improvements in practical situations.
>
> We acknowledge that our theoretical results do not provide a complete mechanistic explanation for why ELRA works in all cases, and we are not claiming to do so. Still, our analysis points to several new and interesting directions for understanding PEFT using Lie-group geometry, such as:
>
> - how constant-speed geodesics can lead to smoother and better-conditioned updates,
> - the link between geodesic energy and a regularizer that naturally fits the geometry,
> - the idea that PSD generators produce spectrally safe updates and help prevent collapse,
> - and a new perspective on PEFT updates, seeing them as controlled paths in $GL(d)$ instead of just local additive changes.
>
> To our knowledge, these ideas provide a geometric way to think about transformer adaptation that is new in the literature and helps explain the stability seen in ELRA-Hyb experiments.

---

> ### Author Response · Authors · 2025-11-26
> **Response to Reviewer Fo85 (Part 3/5)**
>
> # Paper states things that are simply obvious
>
> Thank you for bringing up this point. We agree that statements like “$\exp(G)$ is invertible” and the step from equation (47) to (48) are standard results in linear algebra and Lie group theory. In the revised version of the paper, **we will make necessary changes to make the text more concise.**
>
> It is worth noting that, in our initial submission, we chose to state and prove these facts for two main reasons:
>
> 1. Many PEFT and ML readers may not be familiar with Lie groups, matrix exponentials, or their geometric properties. Without clear statements, the theory—from low-rank generators to invertible updates on $GL(d)$ and the geodesic interpretation—can be hard to follow for non-specialists. By including these short proofs, we make the paper self-contained and help all readers follow the reasoning without needing to look up external Lie-group references.
>
> 2. Although these results are classical, they are important building blocks for our new analysis. For instance, Proposition 3 supports the claim that ELRA preserves the rank of $W_0$, which is key for the conditioning guarantees and stability results. The algebraic steps, such as the move from (47) to (48), are also used directly in deriving the singular-value bounds and geodesic-energy analysis. By keeping these steps explicit, we avoid hidden assumptions and make sure our later theoretical claims are fully transparent.
>
> ---
> # Questions
>
> > Please could you provide a link to the reference: Randlora: A random basis perspective on parameter-efficient fine-tuning In ICLR 2025 by Karimi et al., that you cite in your paper, talk about in your introduction and also talk about on p. 15 of your paper where you explain why your method is better than Randlora by Karimi et al.
>
> This is answered above in the weakness part.
>
> ---
> > Since you spoke about the work Randlora apparently by Karimi et al., why did you not compare against this approach in your experiments? On p. 15 of the paper you explain why your approach is better than Randlora by Karimi et al., yet you don't even compare against their method in your experiments. I found this very strange.
>
> The reason that we did not include this method in our initial experiments is that based on our first inspection, this method does not perform comparable to the stronger PEFT baselines used in our paper. To verify this more concretely, we ran additional experiments during the rebuttal period:
> applied RandLoRA to CLIP-ViT-B/16 on seven image-classification tasks,
> updated Table 3 accordingly,
> and highlighted the new rows in blue.
> The part of the table associated with RandLoRA and some strong baselines along with ELRA variants is reported here:
> | Method| Cars  | DTD   | EuroSAT | GTSRB | RESISC45 | SUN397 | SVHN  | Avg   |
> |--|-|-|--|-|-|-|-|-|
> | RandLoRA        | 83.23 |  77.31|  98.05 | 94.24|  94.37 | 74.19 | 92.97 |  87.77 |
> | LoRA-Pro        | 85.87 | 78.64 | 98.46   | 95.66 | 94.75    | 76.42  | 94.63 | 89.20 |
> | LoRA-GA         | 85.18 | 77.50 | 98.05   | 95.28 | 94.43    | 75.44  | 93.68 | 88.51 |
> | ELRA            | 73.15 | 74.17 | 97.20   | 93.85 | 91.37    | 71.63  | 90.05 | 84.49 |
> | ELRA-PSD        | 78.33 | 75.15 | 97.50   | 94.19 | 93.06    | 74.17  | 93.40 | 86.54 |
> | ELRA-Hyb        | 85.92 | 78.43 | 98.62   | 96.51 | 95.01    | 76.83  | 95.32 | 89.52 |
> | ELRA-Hyb(GA)    | 86.07 | 78.87 | 98.53   | 96.37 | 94.81    | 76.87  | 95.22 | 89.53 |
>
> As shown in the revised table, LoRA-Pro  and LoRA-GA  outperform RandLoRA, and ELRA performs better than all of them. This confirms that RandLoRA's performance is below the more competitive methods we benchmark throughout the paper.

---

> ### Author Response · Authors · 2025-11-26
> **Response to Reviewer Fo85 (Part 4/5)**
>
> > In the experiments you mention (see Section 5) that for all methods the rank = 8 and that for ELRA-PSD the rank = 16. If applying an matrix exponential to the adaptors produces a full-rank matrix that is invertible then this learning is full-rank learning. So shouldn't you be able to use a lower rank say rank 4 and produce better results than the baselines using say rank 16? Even though it might mean your method has less parameters, it would give your method an even better advantage than the others in that it can also use less parameters.
>
> Thank you for your thoughtful question. We answer it in two parts: first, with new empirical evidence from our ablation study, and second, by explaining why lowering the rank does not preserve expressivity even if $\exp(G)$ is full rank.
>
> ## 1. New ablation study
> To test the reviewer’s hypothesis, we ran an ablation study during the rebuttal. The results appear in **Section 6.2 and Figure 2 of the revised paper**.
> We varied the following values:
> $$r \in {4, 8, 16, 32, 64}$$
> We applied these settings to both **LoRA** and **ELRA-Hyb**, and fine-tuned **LLaMA-7B** on several commonsense reasoning benchmarks.
> **Findings:**
> * ELRA-Hyb consistently outperforms LoRA **at every rank**.
> * The **largest gains occur at very low ranks**, which directly answers the reviewer’s question.
> * At $r = 4$: ELRA-Hyb achieves **64.44%**, LoRA **40.10%**.
> * At $r = 8$: ELRA-Hyb **81.83%**, LoRA **41.23%**.
> * The gap narrows at high ranks, but ELRA-Hyb maintains a stable advantage.
> These results show that reducing $r$ is possible, but it does not improve expressivity compared to higher ranks. ELRA-Hyb is more robust than LoRA when the rank is reduced aggressively.
>
> ## 2. Why full-rank adapters do *not* imply full expressivity
> The reviewer’s intuition makes sense: if $\exp(AB)$ is full rank, shouldn’t ELRA work with a very small rank?
> However, it is not the full rank of the matrix that controls expressivity. Instead, it is the dimension of the update manifold.
> ### **(a) ELRA still has only $O(dr)$ trainable degrees of freedom**
> The generator has a rank-$r$ structure:
> $$
> G = \eta AB,\quad
> A\in\mathbb{R}^{d\times r},  B\in\mathbb{R}^{r\times d}.
> $$
> This structure restricts the generator to:
> $$
> \mathcal{L}_r  = {  AB : \operatorname{rank}(AB)\le r }.
> $$
> If we reduce $r$, the following happens:
> * the dimension of $\mathcal{L}_r$ shrinks **quadratically** in $r$,
> * many Lie-algebra directions become inaccessible,
> * and the exponential applies to a much smaller set of generators.
> In summary:
> * Full-rank updates do not imply full expressivity.
> * The generator still lives in a low-dimensional subspace.**
> ### **(b) Exponential never increases the number of degrees of freedom**
> For LoRA:
> $$
> W_{\text{new}} = W_0 + AB \quad \dim = O(dr).
> $$
> For ELRA:
> $$
> W_{\text{new}} = \exp(AB) W_0,\quad \dim = O(dr).
> $$
> The exponential map changes the geometry of the update, but not its dimension.
> Lowering the rank limits the space of possible generators for ELRA just as much as it does for LoRA.
> This explains why $r=4$ performs worse than $r=8$ in our ablation, even though both produce full-rank matrices.
>
> ## 3. Why ELRA-PSD uses $r=16$
> The PSD generator has the form
> $$
> G = AA^\top,
> $$
> This form contains fewer free parameters than a general $AB$ generator.
> To keep the parameter budget fair compared to other baselines, we use $r=16$ for ELRA-PSD so that:
> * the total trainable parameters match the $r=8$ general generators,
> * and the comparison remains balanced across methods.
>
>
> ---
> > In the second paragraph of the introduction you mention (last sentence) "how can we retain parameter efficiency while overcoming rank bottleneck"....
>
> Thank you for raising this point. Ji et al. (ICLR 2025) is conceptually related, and we will add their work to the Related Work section. However, their method does not perform as well as others in our settings, so we did not include it as a baseline.
>
> First, their experimental results do not reach the performance required to compare with the strong PEFT baselines in our paper. ELRA matches or exceeds full fine-tuning performance (see Table 3) while using about 0.5 percent of the original model parameters. The sine-activated adapters in Ji et al. do not achieve this. When their results exceed full fine-tuning accuracy, it occurs only when they use nearly 50 percent of the full fine-tuning parameter count, as shown in Figure 4 of their paper. This indicates their method is not competitive in the low-budget PEFT setting we address.
>
> Second, due to this performance gap, Ji et al. evaluate only against LoRA and DoRA. In our work, DoRA is one of many baselines, and we compare ELRA against a much stronger set of state-of-the-art PEFT methods across large language models, T5, and CLIP.
>
> For these reasons, we did not include Ji et al. in the experimental section. However, we agree their paper is conceptually relevant and will clarify this connection in the revised Related Work section.

---

> ### Author Response · Authors · 2025-11-26
> **Response to Reviewer Fo85 (Part 5/5)**
>
> > On lines 61-65 you speak about the work LieRA by Si et al., and mention that it has a problem in that it limits global feature mixing. What do you mean by this? Could you please give a more detailed explanation of what you mean by that paragraph.
>
> This is answered above in the weaknesses part
>
> ---
> > In section 4.2.2 you mention that ELRA-Hyb interpolates between PSD and general forms starting with a stable geodesic flow and gradually allowing richer dynamics. During optimization how do you see this? The paper doesn't actually show that optimization is better by using ELRA-Hyb.
>
> Thank you for bringing up this question. In Section 4.2.2, we describe ELRA-Hyb to explain how the hybrid schedule leads to more stable optimization in practice, not to claim a formal guarantee.
>
> ### 1. Why ELRA-Hyb improves stability in early optimization
> ELRA-PSD uses a generator of the form
>
> $$
> G_{\text{PSD}} = \eta A A^\top,
> $$
>
> which is symmetric positive semidefinite. As shown in our spectral analysis, the exponential
>
> $$
> \exp(G_{\text{PSD}})
> $$
>
> This has steady and well-controlled effects on the singular values of the layer, which prevents rank collapse and avoids large distortions at the start of training. As a result, early optimization remains stable.
>
> ELRA-Hyb starts in the PSD regime with a large $\alpha$ and then gradually shifts to the general generator as $\alpha$ becomes smaller. This approach keeps the stability benefits of PSD updates early on, while allowing for more expressiveness later.
>
> ### 2. Evidence in the paper: Figure 1
>
> While we do not show optimization trajectories directly, Figure 1 gives indirect empirical support for our explanation. The figure shows that plain LoRA updates often distort the geometry of pretrained features, but PSD-based updates keep the conditioning and smoothness. ELRA-Hyb carries over these stability benefits in early training, helping to avoid unstable or overly aggressive updates that can happen when using a fully general $AB$ generator from the start.
>
> ---
> > Lines 266-290 made no sense to me. What are you trying to say here? Are you saying that optimization depends on the condition number and spectral collapse would be bad?
>
> Thank you for your question. Yes, the main point is that good conditioning helps optimization, while spectral collapse, such as small singular values or very unbalanced spectra, can make training less stable. ELRA-PSD keeps conditioning stable because its PSD generator does not shrink singular values. ELRA-HyB starts training in this stable setting before moving to the more flexible general generator. We have revised this part in the new version of the paper to make this clearer.
>
> ---
> > I would imagine applying the matrix exponential to the adapters would increase the total number of FLOPs. Is this the case?
>
> Thanks for bringing this up. While it might seem like the matrix exponential would increase FLOPs, in ELRA the **cost does not increase in practice** because we never compute $\exp(G)$ as a full $d\times d$ matrix.
> We use the Woodbury-style identity from Proposition 1 in the main paper, which rewrites
>
> $$
> \exp(G)W_0
> $$
>
> with (G = \eta AB) as
>
> $$
> W_0  +
> \eta  A  \phi_1(\eta BA) (BW_0),
> $$
>
> Here, $\phi_1$ is applied only to the **small** $r\times r$ matrix (BA).
>
> This approach reduces the exponential computation to:
> * multiply by (A): (O(dr))
> * multiply by (B): (O(rd))
> * evaluate $\phi_1$ on an $r\times r$ matrix: $O(r^3)$, which is negligible when $r$ is small
> * multiply by $W_0$: this has the same order as LoRA
> So the total cost is:
>
> $$
> \text{FLOPs} = O(drk)
> $$
> This means the total cost is **exactly the same complexity order as standard LoRA**.

---

### Official Review · Reviewer_XNiy · 2025-11-01

**Soundness:** 2
**Presentation:** 3
**Contribution:** 4
**Rating:** 4
**Confidence:** 3

**Summary:**

This paper aim to improve the expressive power of LoRA by replacing the additive update proposed by LoRA into a multiplicative transformation W_new = exp (eta A B) W_0. With exponential, exp(AB) could be full rank. Authors perform some mathematical analysis of the propose method ELRA, and come up with some practical variants ELRA-PSD and ELRA-Hyb for training stablization.

**Strengths:**

- The paper is well written.
    - The problem this paper aim to solve is important.

**Weaknesses:**

- While the idea of making AB full rank is interesting, it’s unclear why this is necessary. With a fixed number of parameters, the overall degrees of freedom remain unchanged. Even if the update is full rank, the representational capacity is still constrained—so which parts of the space become newly accessible?
- I would expect a more fine-grained analysis to clarify this point. For example, when approximating a full update with a low-rank one versus using exp⁡(ηAB)W0−W0, does the latter actually yield a better approximation? Some quantitative analysis, such as approximation error versus number of trainable parameters, would make the argument much stronger.
- Overall, it remains unclear whether the proposed method provides a meaningful advantage over standard low-rank approximations. The experiment results seem good, but I wish this author need to carefully explain why the core idea works.

**Questions:**

See the weakness section

---

> ### Author Response · Authors · 2025-11-26
> **Response to Reviewer XNiy (Part 1/2)**
>
> Thank you for your positive feedback on our motivation, presentation clarity, and contribution. We appreciate that you see the value in improving the expressive power of low-rank adaptation. We are also pleased that the formulation $W_{\text{new}} = \exp(\eta AB), W_0$ and the practical ELRA variants were clear.
> Below are our responses to your concerns.
>
> ---
> # Weakness 1
>
> Thank you for bringing up this issue. The main point is not how many parameters there are (degree of freedom), since both methods use $O(dr)$, but how these parameters are used to change $W_0$. Even though LoRA and ELRA use the same number of parameters, they create very different types of weight updates.
>
> **--->** To better understand this, **consider the following example:** Two DNNs might have the same number of parameters, but one can learn much better because its design allows for more flexible transformations. For instance, a CNN and a fully connected network with the same parameter count do not have the same strengths or possible functions. The key difference is in their structure, not just the number of parameters.
>
> The same idea applies here. ELRA does not add more trainable parameters, but it changes how those parameters affect $W_0$. This changes the way updates can be made, allowing for transformations that LoRA cannot do with the same rank. Here are two points that explain this difference:
>
>
>
> 1.  LoRA produces updates of the form $W_0 + AB$, where $AB$ has rank at most $r$. This restriction confines all changes to a fixed $r$-dimensional row subspace, resulting in the following reachable set: $\mathcal{M}_{\text{LoRA}} = { W_0 + AB }$ is an affine low-rank manifold.
>
> On the other hand, ELRA uses a multiplicative transformation $W = \exp(\eta AB) W_0$, where $\exp(\eta AB)$ is full rank and can be inverted. The set of possible results is: $\mathcal{M}_{\text{ELRA}} = { \exp(G) W_0 }$
>
> This set is part of the full-rank $GL(d)$ orbit of $W_0$, which allows for global changes to its row space.
>
>
>
> 2. With the same $O(dr)$ number of parameters, ELRA allows for transformations that LoRA cannot represent:
>
> * ELRA changes $\text{Row}(W_0)$ to $\text{Row}(W_0)\exp(G)^\top$, which allows for rotations, shears, and different types of scaling of the whole $k$-dimensional subspace. LoRA cannot change this global structure because $AB$ only adds a rank-$r$ part within a fixed subspace.
>
> * Since $\exp(G)$ can be inverted, ELRA keeps both $\text{rank}(W_0)$ and its nullspace unchanged. This avoids the loss of rank that can happen with additive low-rank updates.
>
> * As shown in the paper, $\exp(G)$ keeps the nonzero singular values within certain limits, so the updates stay well-conditioned.
>
> These features describe the **new regions of function space** that ELRA can reach: full-rank, invertible changes to the pretrained layer, not just low-rank additive changes.
>
> ---
> # Weakness 2
> Thank you for highlighting this distinction. ELRA is not meant to approximate any full update $W^\star$ in the Frobenius sense with the same number of parameters. Instead, it aims to make better use of the same low-rank degrees of freedom by shifting from additive low-rank changes to multiplicative full-rank transformations of $W_0$. These are fundamentally different types of approximation problems.
>
> Still, your question is important, and we address it in two parts.
> 1. The expressions
> $$
> W_0 + AB \qquad \text{and} \qquad \exp(\eta AB)W_0
> $$
> represent two different types of functions:
> * $W_0 + AB$ approximates $W^\star$ through *additive* low-rank variation.
> * $\exp(\eta AB)W_0$ approximates $W^\star$ through *full-rank reparameterization* of the row space of $W_0$.
> Since $\exp(\eta AB)$ is invertible, $\exp(\eta AB)W_0$ stays on the $GL(d)$ orbit of $W_0$, which forms a highly structured manifold. As a result:
> * ELRA cannot and is not designed to approximate *all* matrices $W^\star$.
> * But within the orbit $GL(d)\cdot W_0$, ELRA provides **strictly stronger expressivity** per parameter, because it modifies the *entire* row space of $W_0$, instead of adding rank-$r$ corrections.
> So, the nature of the approximation problem is different. ELRA improves how low-rank parameters influence $W_0$ instead of trying to match full updates.
> 2. While our theory is based on geometry rather than approximation, the experimental results still answer the reviewer’s question with data:
> * ELRA and LoRA variants use the **same rank $r$ and the same number of trainable parameters**.
> * We consistently see lower errors, such as lower perplexity, higher accuracy, and lower validation loss, when using the same parameter budgets.
> * In some cases, ELRA even outperforms full fine-tuning, which would not happen if $\exp(\eta AB)W_0$ were just a worse approximation method.
> These results provide strong evidence for the quality of the approximation: with the same number of parameters, $\exp(\eta AB)W_0$ gives better performance than $W_0 + AB$.

---

> > ### Author Response · Authors · 2025-11-26
> > **Response to Reviewer XNiy (Part 2/2)**
> >
> > # Weakness 3
> >
> > Thank you for bringing up this important point. We have already addressed the main ideas behind ELRA in my earlier responses to your questions:
> > * why multiplicative full-rank transformations with the same $O(dr)$ parameters access a fundamentally different update manifold than $W_0 + AB$,
> > * why this difference is structural rather than dimensional,
> > * and how this leads to changes in the entire row space of $W_0$, not just an $r$-dimensional slice, providing the concrete expressivity advantage.
> >
> >
> > Lastly, we mention two important points:
> >
> >
> > 1. ELRA is not meant to approximate any possible full update, so comparing $\exp(\eta AB)W_0 - W_0$ to a full-rank target, as you would with $AB$, does not make sense. The key comparison should be between the two update manifolds, not how well they can match any matrix.
> > 2. The experiments already show the main point: with the same number of trainable parameters, ELRA consistently gets lower error, higher accuracy, and better perplexity. This means the geometric argument does lead to better results in practice.

---

### Note · Program_Chairs · 2026-01-17
**Submission Desk Rejected by Program Chairs**

The following references in this submission do not refer to real documents and/or have major errors in bibliographic information:

 Yichong Zhang, Junnan Zhao, Xitong Xu, Min Cheng, Xiang Wang, and Shuai Wang. Abba: Adaptive bitwise binarization of attention for efficient transformers. arXiv preprint arXiv:2403.02478, 2024